# KERNELFUSION: ZERO-SHOT BLIND SUPER-RESOLUTION VIA PATCH DIFFUSION

**Oliver Heinimann** [*, 1]     **Tal Zimbalist** [*, 1]     **Assaf Shocher** [2]     **Michal Irani** [1]

[1] Weizmann Institute of Science     [2] Technion

{oliver.heinimann}@weizmann.ac.il

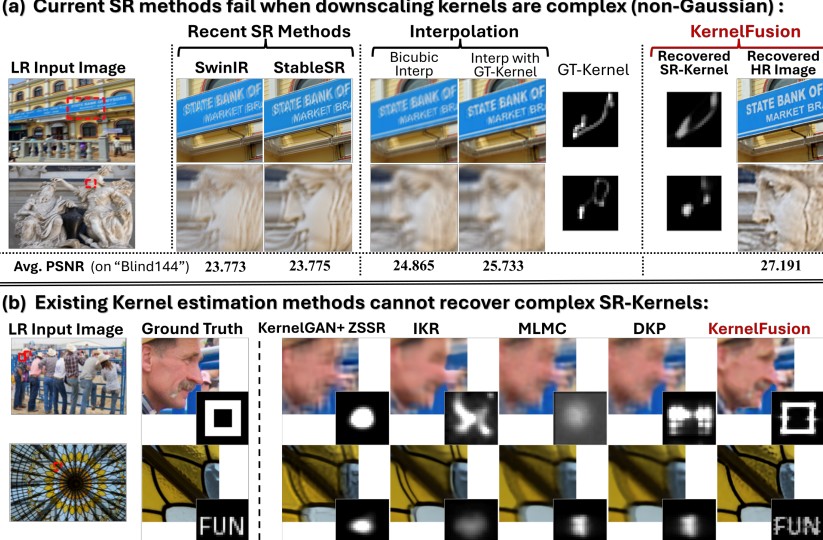

Figure 1: **The importance of an accurate SR-Kernel.** (A) SotA SR-methods fail on complex downscaling kernels, performing even worse than interpolation on such kernels. (B) Existing SR-kernel estimation methods cannot handle complex kernels. KernelFusion is the only method capable of estimating arbitrary SR-kernels.

## ABSTRACT

Traditional super-resolution (SR) methods assume an "ideal" downscaling SR-kernel (e.g., bicubic downscaling) between the high-resolution (HR) image and the low-resolution (LR) image. Such methods fail once the LR images are generated differently. Current blind-SR methods aim to remove this assumption, but are still fundamentally restricted to rather simplistic downscaling SR-kernels (e.g., anisotropic Gaussian kernels), and fail on more complex (out of distribution) downscaling degradations. However, using the correct SR-kernel is often more important than using a sophisticated SR algorithm. In "KernelFusion", we introduce a zero-shot diffusion-based method that uses an unrestricted kernel. Our method recovers the unique image-specific SR-kernel directly from the LR input image, while simultaneously recovering its corresponding HR image. KernelFusion exploits the principle that the correct SR-kernel is the one that maximizes patch similarity across different scales of the LR image. We first train an image-specific patch-based diffusion model on the single LR input image, capturing its unique internal patch statistics. We then reconstruct a larger HR image with the same learned patch distribution, while simultaneously recovering the correct downscaling SR-kernel that maintains this cross-scale relation between the HR and LR images. Empirical results demonstrate that KernelFusion handles complex downscaling degradations where existing Blind-SR methods fail, achieving robust kernel recovery and superior SR quality. By breaking free from predefined kernel assumptions and training distributions, KernelFusion establishes a new paradigm of zero-shot Blind-SR that can handle unrestricted, image-specific kernels previously thought impossible. *See project's webpage for additional information.*

---

[*]These authors contributed equally.

## 1 INTRODUCTION

Super-resolution (SR) is an inverse problem of recovering a high-resolution (HR) image from its low-resolution (LR) counterpart, given by:

$$I_{LR} = (I_{HR} * k_s) \downarrow_s, \qquad (1)$$

where $k_s$ is the downscaling kernel (also known as SR kernel) and $\downarrow_s$ denotes subsampling by a scale factor $s$. Traditional SR methods have achieved impressive results Dong et al. (2015); Lim et al. (2017); Kim et al. (2016); Zhang et al. (2018b;c); Saharia et al. (2022); Li et al. (2024) under the assumption that $k_s$ is a global, known kernel (e.g. bicubic with antialiasing), but this is rarely the case. The SR-kernel tends to be *image-specific*; it is affected not only by sensor optics, but also by camera motion, subtle hand movements, and other factors. Evidently, these methods perform poorly in any scenario other than synthetic data specifically created using the assumed kernel. In fact, it was shown Levin et al. (2009); Efrat et al. (2013), that the *accuracy of the SR-kernel is often more critical for obtaining good SR*, than the image prior or the choice of SR algorithm used.

Blind-SR methods have emerged to address this limitation. Some approaches aim to explicitly estimate the unknown kernel (e.g., Michaeli & Irani (2013); Bell-Kligler et al. (2019); Xia et al. (2024)), whereas others represent the SR-kernel implicitly Gu et al. (2019); Luo et al. (2023); Huang et al. (2020); Kim et al. (2021); Ates et al. (2023), or aim to design networks that are robust to kernel variations (e.g., Liang et al. (2021); Wang et al. (2024b;a); Luo et al. (2022; 2025); Lin et al. (2024)). However, existing Blind-SR methods are fundamentally limited: They can only super-resolve well LR images which were downscaled by simple, low-pass-filter kernels (e.g., (an)isotropic Gaussians), and fail on more complex downscaling kernels, which are outside their training distribution. In fact, *for LR images obtained by non-Gaussian downscaling kernels, SOTA Blind-SR methods perform worse than simple interpolation.* (see Fig. 1a and Sec. 3).

In "KernelFusion", we introduce a zero-shot diffusion-based method that makes no assumptions (explicitly or implicitly) about the downscaling kernel, other than the kernel being global. Our method recovers the unique image-specific SR-kernel directly from the LR input image, while *simultaneously* recovering its corresponding HR image. KernelFusion exploits the principle (presented by Michaeli & Irani (2013) and used in Bell-Kligler et al. (2019)), that the correct SR-kernel is the one that also maximizes patch similarity across different scales of the LR image. More specifically, we first train an *image-specific* patch-based diffusion model on the single LR input image, capturing its unique internal patch statistics. We then reconstruct the larger HR image during the reverse diffusion process, enforcing the same patch distribution, while simultaneously estimating downscaling SR-kernel. While existing methods excel at handling Gaussian kernels, empirical results show that KernelFusion succeeds in recovering valid kernels and reconstructions under complex degradations, scenarios where prior Blind-SR methods tend to break down.

The ability of KernelFusion to handle complex downscaling kernels (where all previous SR methods fail), stems from the following critical design choices:

**1.** Being a zero-shot estimation method which trains *internally* on the LR input image only, KernelFusion is not bound by any external training distribution, hence can handle any type of downscaling kernel. There is no notion of "out-of-distribution" kernels, which *externally-trained* Blind-SR methods suffer from (see Fig. 1a).

**2.** Previous zero-shot SR-kernel methods Michaeli & Irani (2013); Bell-Kligler et al. (2019) estimated the kernel only, requiring a separate independent SR algorithm to super-resolve the LR image with their recovered kernel (e.g., using ZSSR Shocher et al. (2018b) or SRMD Zhang et al. (2018a), which can receive a user-specified kernel). Such a 2-step process suffers from accumulated errors and inconsistencies between the estimated kernel and the estimated HR image. In contrast, KernelFusion *simultaneously* estimates both the SR-kernel and the HR image *in a consistent manner*.

**3.** The explicit kernel estimation methods of Bell-Kligler et al. (2019); Xia et al. (2024); Ates et al. (2023); Yang et al. (2024b) seem to recover well only specific types of kernels (Gaussians and motion lines). We suspect that this limitation stems from the implicit-bias of the CNN and MLP architectures, which tend to produce smooth outputs (as also observed in Tancik et al. (2020)). In contrast, the kernel-estimation component in KernelFusion employs an *Implicit Neural Representation* (INR) architecture, which recovers complex *non-smooth* downscaling SR-kernels (see Fig. 1).

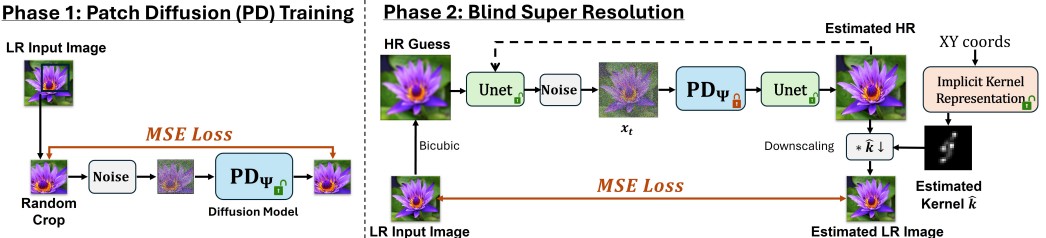

Figure 2: **Method Overview.** Our approach consists of 2 stages: **Phase 1**: We train a diffusion model (PD) to learn the patch distribution of a single image. **Phase 2**: We perform blind SR and kernel estimation simultaneously. In particular, we use the trained PD to shift the HR guess toward the patch distribution of the LR input. A refinement U-Net and an implicit kernel representation model are trained jointly under a consistency loss, ensuring that convolving the estimated HR image with the learned kernel reproduces the original LR image.

Our aim is not to deliver a production-ready Blind-SR system, but to establish feasibility of unrestricted kernel estimation from a single LR image while simultaneously recovering the HR image. By breaking free from predefined kernel assumptions and training distributions, KernelFusion pushes Blind-SR into a new, assumption-free paradigm – handling out-of-distribution data and downscaling kernels which were previously out of reach.

- *KernelFusion* is the first deep Blind-SR method able to recover arbitrary downscaling kernels.
- Since *KernelFusion* trains on the LR input image, there is *no notion* of "out-of-distribution" data. It is trained to adapt to the *image-specific* data and downscaling kernel.
- *KernelFusion* provides state-of-the-art SR results on challenging *out-of-distribution* data, where leading SR methods fail (while being competitive within distribution).

## 2 RELATED WORK

We first describe three main types of Blind-SR Liu et al. (2022). We then review diffusion models and their use for inverse problems, which is related to our method.

**Blind-SR trained on synthetic degradations:** These methods train on LR images generated by synthetic degradations from a *predefined* distribution of degradations. When applied to data close to that distribution they achieve visually pleasing results. These include SwinIR Liang et al. (2021), Real-ESRGAN Wang et al. (2021) and many more Conde et al. (2022); Zhang et al. (2021); Jo et al. (2021); Luo et al. (2022); Lin et al. (2024); Sun et al. (2024); Wu et al. (2024b); Wei et al. (2020); Yang et al. (2024a); Zhang et al. (2024); Wu et al. (2024a); Chen et al. (2025). Such methods are restricted by their training distribution, thus fail on LR images generated by out-of-distribution downscaling kernels (e.g., non-Gaussian kernels).

**Blind-SR with latent kernel representation:** Other methods represent degradations by a latent features vector. These methods perform SR and refinement of the degradation features, often based on alternating SR and latent kernel estimation. This was first shown by IKC Gu et al. (2019), later improved by unfolding this alternation with DAN Huang et al. (2020); Luo et al. (2023) and others Luo et al. (2022); Kim et al. (2021); Zhang et al. (2020); Sohl-Dickstein et al. (2015); Mehta et al. (2025). The kernel representation is data-driven, hence also based on the synthetic degradations used at training. These methods too, fail on images downscaled by kernels out of their training distribution.

**Blind SR-Kernel Estimation:** Acknowledging the importance of an accurate SR-kernel, some approaches aim to explicitly estimate the unknown SR-kernel directly from the LR image (e.g., Michaeli & Irani (2013); Bell-Kligler et al. (2019); Xia et al. (2024); Ates et al. (2023); Yang et al. (2024b); Tao et al. (2021)). Notably, Michaeli & Irani (2013) was the first to observe that the optimal SR-kernel is the one that maximizes the similarity of small patches *across* different scales of the LR image, and accordingly used cross-scale patch nearest-neighbors to estimate the SR-kernel. KernelGAN Bell-Kligler et al. (2019) further used this principle within deep learning, showing that

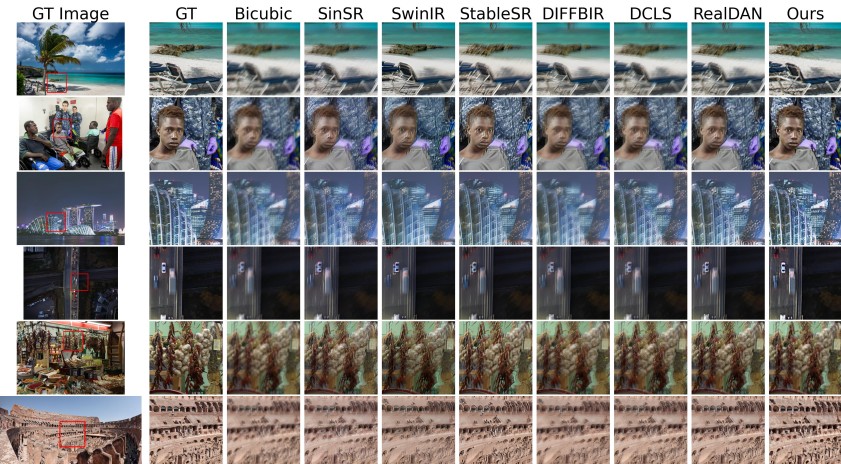

Figure 3: **Blind-SR comparison** on the DIV2KFK dataset (4× SR). Each row relates to a different LR image from DIV2KFK, while each column shows the output of a different method. Notably, our method reduces doubling artifacts in structured patterns (e.g., aerial road scene, 4th row), demonstrating its effectiveness in restoring fine details and mitigating motion effects.

the SR-kernel can be estimated by training an image-specific GAN on the LR image. However, the zero-shot methods of Michaeli & Irani (2013); Bell-Kligler et al. (2019) are pure kernel estimation methods. They do not perform any SR, hence require a separate followup algorithm to perform the SR step on the LR image (e.g., using ZSSR Shocher et al. (2018b) or SRMD Zhang et al. (2018a)), which can receive a user-specified kernel as an input. This restricts their applicability. Moreover, the explicit kernel estimation methods of Bell-Kligler et al. (2019); Xia et al. (2024); Ates et al. (2023); Yang et al. (2024b) can only recover well specific types of kernels (Gaussians and motion lines – see examples in Fig. 1).

**Diffusion Models and Inverse Problems:** Diffusion probabilistic models Sohl-Dickstein et al. (2015); Ho et al. (2020) have become a powerful tool for modeling complex image distributions. More recently, under the Deep Internal Learning regime Shocher et al. (2018b); Gandelsman et al. (2019); Ulyanov et al. (2018); Shocher et al. (2018a); Shaham et al. (2019); Granot et al. (2022) diffusion-based approaches have been adapted for the single-image setting Nikankin et al. (2023); Wang et al. (2022); Kleiner et al. (2023). Additionally, diffusion models have shown promise in solving inverse problems such as deblurring and super-resolution Kawar et al. (2022); Chung et al. (2022). Recent works further enhance these approaches by incorporating data-consistency constraints via null-space projections Wang et al. (2023) or by adding back-projection steps Hui et al. (2024). Plug-and-play (PnP) adaptions to diffusion or flow matching, such as Zhu et al. (2023); Martin et al. (2024), use the model as a denoiser in an iterative reconstruction algorithm. However, these methods do not cover the case of blind SR. Additional work on patch-based diffusion models Altekrüger et al. (2023); Hu et al. (2024a;b) further exploits local image statistics for improved detail recovery. Due to their stochastic nature, diffusion models struggle to adhere to measurement constraints. Moreover, they may exhibit a distribution mismatch between the training data and the observed measurements, necessitating careful adaptation Hu et al. (2024b). A related line of work leverages diffusion priors for kernel estimation in inverse problems, most notably BlindDPS Chung et al. (2023), which tackles blind deblurring and imaging through turbulence. While conceptually related to KernelFusion, BlindDPS operates in a different regime, relying on pre-trained diffusion models over images and kernels trained on large synthetic blur datasets, whereas KernelFusion is entirely zero-shot.

## 3 THE IMPORTANCE OF AN ACCURATE SR-KERNEL

The accuracy of the SR-kernel is critical for achieving high-fidelity HR image reconstruction, often playing a more crucial role than the image prior or SR method itself Efrat et al. (2013); Levin et al. (2009). The SR process fundamentally relies on inverting the degradation introduced by downsam-

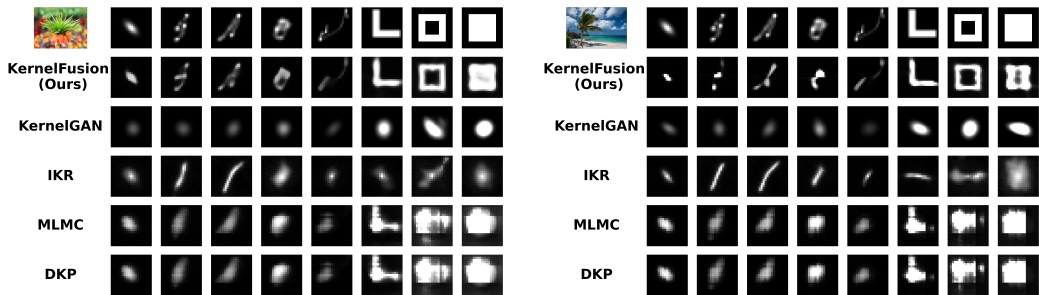

Figure 4: **Comparison of estimated kernels from different Blind-SR methods.** The top row represents the ground-truth (GT) degradation kernels, while each subsequent row corresponds to the estimated kernels from different SR methods, including our approach, KernelGAN, IKR, MLMC, and DKP. Our method demonstrates superior flexibility in recovering complex, non-Gaussian degradations, accurately capturing kernel structures across a diverse range of degradations.

pling, which is dictated by the underlying kernel. If the kernel is inaccurate or out of distribution, even advanced SR models risk producing artifacts and unrealistic results.

This property is shown in Fig. 1: Two leading algorithms (SwinIR & StableSR), both trained with blind SR degradations, fail to accurately upscale LR images which were downscaled by non-Gaussian kernels. In contrast, *simple interpolation* leads to remarkably better visual results. We further confirm this observation quantitatively on 2 datasets with hundreds of LR images downscaled by a variety of *non*-gaussian kernels ("Blind144" & "DIV2KFK" – see Sec. 5. These results are summarized in Table 2, and a sample is displayed here (in Table 1), showing that even SotA Blind-SR methods (DPSR Zhang et al. (2019), DCLS Luo et al. (2022)) perform worse than simple bicubic interpolation on non-gaussian kernels (which are outside their training distribution). Moreover, applying more *sophisticated interpolation*[0] with the ground-truth (GT) kernel provides an additional large improvement (by +1dB) over SotA Blind-SR methods on such kernels.

## 4 METHOD: "KERNELFUSION" FOR BLIND-SR

Our method builds on the principle that the correct SR kernel is the one that best preserves the image's patch distribution across scales. In a first step, a patch-based diffusion model is trained on the LR input image, learning its patch distribution. In a second step, we perform super resolution by iteratively improving our HR guess and estimating the SR-kernel during the reverse diffusion process. Fig. 2 provides an overview over our approach.

**Phase 1: Training Patch-Diffusion (PD):** Our Patch-Diffusion (PD) model aims to learn the patch distribution of the LR input image $I_{LR}$. As diffusion models are excellent distribution learn-

Table 1: SotA Blind-SR methods perform *worse than interpolation* on LR images downscaled by non-Gaussian kernels.

| Method | Blind144 | DIV2KFK |
|---|---|---|
| Bicubic | 24.865 | 24.101 |
| GT kernel | 25.733 | 25.057 |
| DPSR | 24.824 | 23.997 |
| DCLS-SR | 24.808 | 23.886 |

ers, we base our model on the standard denoising probabilistic diffusion model (DDPM) Ho et al. (2020), predicting a velocity $v$ (inspired by Salimans & Ho (2022)). We motivate the $v$ parametrization by the increased stability i.e. at few sampling steps. We train it to denoise solely the single LR input image noised with various noise magnitudes, according to a standard DDPM schedule. More specifically, the input is a diffused $I_{LR}$ to random time step $t \in [1, T]$ while the target is the velocity (from which one can easily derive the clean original input image using a closed form):

$$\Psi = argmin_\psi \left\| PD_\psi(x_t) - v_t \right\|_2^2, \text{ where } x_t = \sqrt{\bar{\alpha}_t}x_0 + \sqrt{1-\bar{\alpha}_t}\epsilon, \quad v_t = \sqrt{\bar{\alpha}_t}\epsilon - \sqrt{1-\bar{\alpha}_t}x_0, \Psi \text{ are}$$

the parameters of our PD model, $\epsilon$ a standard normal white noise, $x_0 = I_{LR}$ the LR input, $\alpha_t$ the standard DDPM coefficient, and $\bar{\alpha}_t = \prod_{s=1}^{t} \alpha_t$.

---

[0]Interpolation with a kernel is obtained by backprojection Irani & Peleg (1991) with the estimated Pseudo-Inverse Moore (1920); Penrose (1955) of the kernel.

Learning the patch distribution of a single image requires some adjustments: We took inspiration from Nikankin et al. (2023) using pure CNN without any global layers (such as attention). However, since our method is based on much smaller patches, the receptive field needs to be restricted even further. Thus we exchange the backbone with a simple convolutional network with no strides, inducing a *theoretical* receptive field of $15 \times 15$ pixels (and practically much smaller). This allows training PD on random $64 \times 64$ image crops (see Phase 1 in Fig. 2). Note the difference between the larger $64 \times 64$ image "*crops*", and the smaller "*patches*" whose distribution is being learned (and whose size is determined by the small PD receptive field). Each random $64 \times 64$ image crop serves as a *batch* of thousands of small image patches. Further design choices, hyperparameters, and implementation details are found in *Implementation Details*.

**Phase 2: Reverse Diffusion at High Resolution:** Once PD is trained, it has learned the patch distribution of the LR input image. Next, we perform *simultaneous* SR and kernel estimation, based on the reversed sampling process of PD [1]. This constrains the produced HR image to have the same patch distribution as $I_{LR}$ (which it should have Michaeli & Irani (2013)). This combined approach has a major benefit: A better HR prediction $\hat{x}_0$ leads to an improved kernel prediction, and vice versa. The right part of Fig. 2 (labeled as Fig. 2R below) summarizes the main components used in the reversed sampling process while Algorithm 1 in the appendix, provides pseudo-code of the approach, which is further detailed below. Our method takes as an initial input a bicubically upscaled version of the LR image $I_{LR}$ (top left Fig. 2R). This bicubic guess is noised by $T_{nd}$ steps using the diffusion noise schedule ("Noise" in Fig. 2R). Subsequently, we would like to optimize $\hat{x}_0$ at each timestep $t$, such that the downscaled version $\hat{x}_0 \downarrow_s$ (using the kernel estimated by the INR, Fig. 2R on the right) is consistent with the input image $I_{LR}$ (bottom row of Fig. 2R). Instead of directly optimizing $\hat{x}_0$, we chose to implicitly optimize $\hat{x}_0$ via a U-Net (see right "UNet" in Fig. 2R). The U-Net (inspired by DIP Ulyanov et al. (2018)) imposes a *global* image prior on the output. The PD step used to predict $\hat{x}_0$ alone does not inherently preserve global structure, especially at high $t$, due to the $15 \times 15$ local receptive field of PD (see Appendix D further details). We solve this problem by applying the (same) U-Net twice: First, we apply the U-Net to $x_0$ from the previous timestep $t + 1$ and reconstruct the required $x_t$ from it. Second, we apply the U-Net after denoising $x_t$ using our patch-diffusion model $PD_\psi$ to the predicted $x_0$ at timestep $t$. We refer to Appendix C.2 for more details and intuition. Optimizing the U-Net using our LR consistency loss is hence key to maintain global structure. This setup enables joint training of both the U-Net and the kernel estimation network (see *Kernel Estimation using INR*) with a single loss. Since both networks are trained from scratch, we apply $n_{iter}$ gradient steps at each timestep $t$. The networks are iteratively refined at every $t$, leveraging their gradual improvements to enhance the prediction of $\hat{x}_0$.

**Kernel Estimation using INR:** Estimating a SR-kernel means solving for $k_s$ in Eq. (1). Instead of directly solving for $k_s$, we chose to represent the kernel via an Implicit Neural Representation (INR) network, which allows us to represent the kernel *continuously* Shocher et al. (2020), while controlling its level of regularization. Specifically, we took inspiration by the SIREN architecture Sitzmann et al. (2020) which is known for its ability to also represent high frequency functions. Specifically, the sinusoidal activations enable the network to capture fine-grained structures without introducing over-smoothing, which is critical for accurately estimating complex downscaling kernels $k_s$. See A.4 for more details.

**Consistency Loss:** In Phase 2, our method relies on a MSE consistency loss (see Eq. (2) to enforce a pixel-wise alignment between the input LR image $I_{LR}$ and the estimated LR image $(\hat{x}_0 * \hat{k}_s) \downarrow_s$. This prevents predicting hallucinated structures not supported by the $I_{LR}$.

$$\mathcal{L}_{\text{cons}} = \text{MSE}\Big( I_{LR}, \Big( \hat{x}_0 * \hat{k}_s \Big) \downarrow_s \Big) \tag{2}$$

**Implementation Details:** To ensure a small receptive field of PD, we exchanged the global-receptive-field U-net of the original DDPM architecture with a small, fully convolutional neural net. We use one block consisting of two $3 \times 3$ convolutions, followed by five blocks $3 \times 3 + 1 \times 1$ convolutions. This results in a theoretical receptive field of $15 \times 15$ pixels. The diffusion model is trained with $T = 1000$ time steps. For more details see appendix A.

---

[1]Note that during Phase 2, the PD network is frozen with gradients allowed to propagate through it. We indicate this by a lock symbol in Fig. 2

Table 2: **Quantitative results on 4× SR across Blind-SR datasets.** PSNR/SSIM for a representative set of methods; the two best results per dataset are shown in **bold** and underline. On the 2 non-Gaussian datasets (DIV2KFK, Blind144), KernelFusion is able to recover consistent kernels and reconstructions where many prior methods struggle, while bicubic interpolation serves as a surprisingly strong reference. On the Gaussian DIV2KRK dataset, performance remains comparable to methods explicitly trained for such kernels. *__Comment__*: The reported PSNRs are after excluding an image boundary of 5%, due to boundary effects.

| Method | Blind144 | | DIV2KRK | | DIV2KFK | |
| --- | --- | --- | --- | --- | --- | --- |
| | PSNR↑ | SSIM↑ | PSNR↑ | SSIM↑ | PSNR↑ | SSIM↑ |
| Bicubic | 24.865 | 0.637 | 25.075 | 0.671 | 24.101 | 0.639 |
| SinSR Wang et al. (2024b) | 23.587 | 0.582 | 25.360 | 0.680 | 22.887 | 0.585 |
| StableSR Wang et al. (2024a) | 23.775 | 0.625 | 25.262 | 0.709 | 23.077 | 0.630 |
| OSEDiff Wu et al. (2024a) | 22.841 | 0.585 | 23.867 | 0.658 | 22.359 | 0.602 |
| SwinIR Liang et al. (2021) | 23.773 | 0.616 | 25.139 | 0.699 | 23.070 | 0.620 |
| DPSR Zhang et al. (2019) | 24.824 | 0.637 | 25.317 | 0.682 | 23.977 | 0.637 |
| IKR Ates et al. (2023) | 24.113 | 0.630 | 23.906 | 0.667 | 23.352 | 0.627 |
| RealDAN Luo et al. (2023) | 24.624 | 0.638 | 26.870 | 0.745 | 23.941 | 0.644 |
| RealDAN gan Luo et al. (2023) | 23.998 | 0.619 | 26.057 | 0.730 | 23.439 | 0.628 |
| RealDAN spec. Luo et al. (2023) | - | - | 27.821 | 0.775 | - | - |
| DCLS-SR Luo et al. (2022) | 24.808 | 0.633 | 27.150 | 0.748 | 23.886 | 0.634 |
| DiffBIR Lin et al. (2024) | 24.259 | 0.599 | 25.431 | 0.668 | 23.546 | 0.606 |
| DRAT Chen et al. (2025) | 24.747 | 0.631 | **27.953** | **0.779** | 23.824 | 0.631 |
| DKP Yang et al. (2024b) | 23.431 | 0.612 | 23.127 | 0.629 | 22.531 | 0.603 |
| MLMC Xia et al. (2024) | 23.430 | 0.612 | 23.122 | 0.629 | 22.535 | 0.604 |
| KernelGAN+ZSSR Shocher et al. (2018b) | 24.529 | 0.633 | 25.895 | 0.703 | 23.617 | 0.629 |
| Real-ESRGAN Wang et al. (2021) | 23.646 | 0.610 | 24.323 | 0.678 | 23.003 | 0.617 |
| *KernelFusion (ours)* | **27.191** | **0.719** | 26.761 | 0.715 | **26.426** | **0.720** |

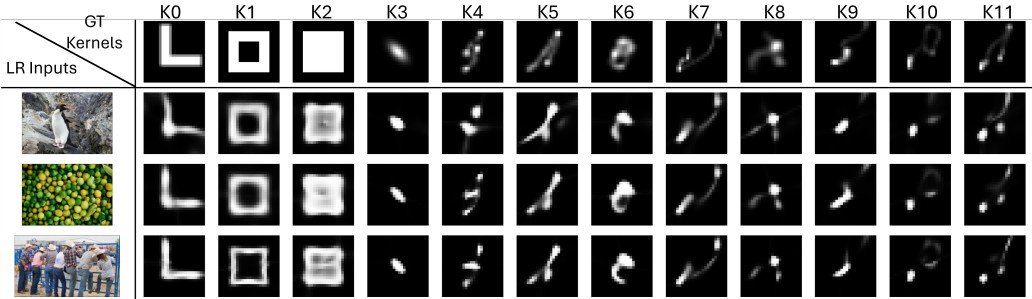

Figure 5: **Kernel estimation results on Blind144**. The top row displays the 12 ground-truth (GT) degradation kernels, including real-world motion blur kernels from Levin et al. (2009), an anisotropic Gaussian kernel, and three synthetic non-natural kernels (L-shape, empty square, and filled square). The subsequent rows show our method's estimated kernels for each of the 12 kernels applied to 3 sample images of the DIV2K validation set. Our approach successfully captures a diverse range of degradations, including complex structured kernels, demonstrating its robustness and adaptability in blind SR kernel estimation.

## 5 RESULTS

Our evaluation targets unrestricted kernels through (i) complex non-Gaussian kernels, and (ii) case studies highlighting failure modes of prior methods. Further, we evaluate on a Gaussian baseline.

**DIV2KRK Bell-Kligler et al. (2019):** LR images were obtained by downscaling each of the 100 images in the DIV2K validation set Agustsson & Timofte (2017), with a different random *anisotropic Gaussian kernel* of varying sizes and orientations.

**DIV2KFK:** Inspired by DIV2KRK, we created a new dataset based on the 8 real-life downscaling kernels measured by Levin Levin et al. (2009), which were induced by small camera jitter during the shutter exposure time. We call this dataset DIV2KFK (DIV2K-Fancy-Kernels). In order to have reasonable blur levels for 4× SR, we resize the original kernels to size 24×24. Each image of the DIV2K validation set is convolved with a randomly selected kernel and sampled by a scale of 4.

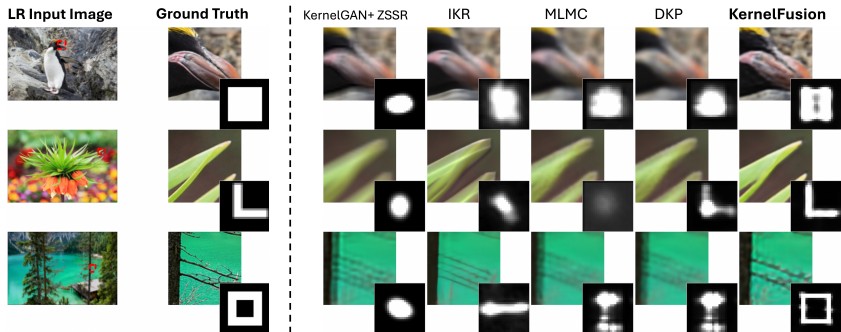

Figure 6: **Examples of *explicit* SR-kernel estimation and SR under extreme downscaling (Blind144).**

**Blind144:** Two factors affect the Blind-SR challenge: the image and the downscaling kernel. To further analyze and better understand SR results, we created the controlled Blind144 dataset. It is organized as a 12×12 matrix comprising 12 images and 12 kernels. This setup allows us to examine the results of a single kernel across various images, as well as the outcomes of a single image across different kernels, thereby clarifying whether specific effects and behaviors stem from the image or the kernel. The images used are the first 12 images from DIV2K. For the selection of kernels we used the 8 empirical kernels as in DIV2KFK, and in addition 3 extreme stress-testing kernels and one anisotropic Gaussian. The purpose of the Gaussian kernel is to compare quality of results on the same image for in-distribution vs. out-of-distribution kernels. The 3 extreme kernels contain an "L" shaped kernel, a full square with sharp edges, and an empty square. These allow stress-testing of kernel extraction. The full set of kernels is shown in the top row of Fig. 5.

**Empirical Evaluation of SR:** Since our goal is to assess accurate reconstruction with respect to ground truth, we use alignment-sensitive metrics such as peak signal-to-noise ratio (PSNR) and structural similarity index measurement (SSIM) Wang et al. (2004). Perceptual metrics, while useful for visual quality, are less suited for measuring pixel-level fidelity due to alignment insensitivity. We orient our evaluation on the procedure of Kim et al. (2016): The evaluation is executed on the luminance channel (YCBCR space). Table 2 compares KernelFusion with representative SotA competitors. Our method is able to recover accurate kernels and reconstructions on challenging non-Gaussian datasets (DIV2KFK and Blind144), where prior blind-SR methods fail. On the Gaussian DIV2RK dataset, where existing approaches are specifically tailored to, our results remain comparable. It is worth noting that a simple bicubic upscaling surpasses all SotA methods in PSNR on DIV2KFK & Blind144, indicating that these methods completely fail adapting to *out-of-distribution* downscaling.

Fig. 3 showcases SR results on the DIV2KFK dataset, comparing our method against SotA approaches. Our method demonstrates superior reconstruction quality, particularly in challenging regions such as text, structured patterns, and natural textures. Notably, in Fig. 1, row 1, which contains the State Bank of Mysore sign, most competing methods struggle to recover clear and legible text, often introducing excessive blurring or ghosting artifacts, whereas our method produces sharper, more readable characters. A similar trend is observed in the aerial road scene (Fig. 3, row 4), where other methods introduce noticeable doubling artifacts (misaligned white car), which are significantly reduced in our results. Furthermore, our approach maintains a balance between sharpness and natural texture preservation in human faces and complex textures, avoiding the over-sharpening and aliasing effects present in some models (e.g., RealDAN, DIFFBIR). These results demonstrate the robustness of our approach in handling diverse realistic degradations as in the DIV2KFK dataset.

**Kernel Evaluation:** To assess the effectiveness of different blind-SR methods in kernel estimation, we visualize the predicted SR-kernels in Fig. 4. The top row represents the ground-truth (GT) kernels, while subsequent rows correspond to the estimated kernels from various blind-SR methods that explicitly estimate the SR-kernel (including ours). KernelGAN demonstrates a strong bias toward Gaussian-like kernels, failing to accurately capture non-Gaussian degradations. IKR Ates et al. (2023), which was trained with a mixture of random Gaussian and motion blur kernels, exhibits improved performance on motion kernels but struggles to generalize beyond its training distribution, often producing elongated blur patterns. Both MLMC Xia et al. (2024) and DKP Yang et al. (2024b)

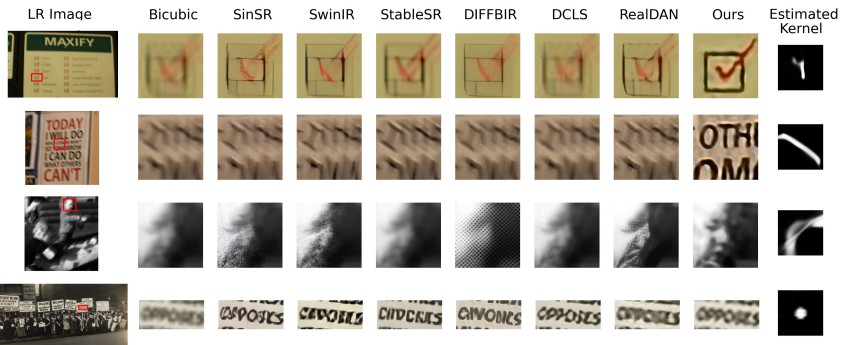

Figure 7: **SR×4 on diverse *real-world* images.** Each real input image is taken from a different source. Top-to-Bottom: (i) Image from Lai et al. (2016), (ii) A photo captured with a DSLR camera *without* optical stabilization. (iii) Image from Levin et al. (2009), (iv) An old historical photo downloaded from the internet. Zoomed-in regions (highlighted in red) show that our method better recovers fine structures, text, and textures compared to SotA Blind-SR methods. For example, our method clearly recovers the word "OPPOSES" in the old historical photo (bottom), which remains unreadable in most competing methods.

leverage meta-learning and Markov Chain Monte Carlo (MCMC) simulations to infer kernel priors without explicit assumptions. As a result, they are more flexible in capturing diverse degradation patterns. However, despite their adaptive nature, their estimated kernels still exhibit noticeable deviations from the GT, particularly in the case of extreme degradations. Unlike prior methods, our approach does not impose any priors on the kernel shape, allowing it to accurately recover challenging degradations. Fig. 5 presents our estimated kernels across samples from the Blind144. As observed, our approach is capable of recovering a diverse range of kernel structures, demonstrating the flexibility of our kernel estimation process. The entire collection of 144 kernels recovered by KernelFusion from the Blind144 dataset can be found in the appendix, Fig. 10. Fig. 6 shows some more SR and kernel extraction comparison for extreme kernels.

**Ablations:** We ablate the reverse diffusion process described in Phase 2 (see Fig. 2) to identify the individual component contributions (see Table 3): **(i) DIP:** The UNet gets pure noise as input and is optimized with the consistency loss on the downscaled prediction Eq. 2, together with the INR. This corresponds to DIP Ulyanov et al. (2018). We observe that the UNet alone can already account for a certain adjustment of the patch distribution, while making use of the powerful INR. **(ii) UNet only:** Here, the UNET and the INR are optimized within the full reverse diffusion framework, starting from a partially noised image. At each t, the UNet and INR are optimized for 20 iterations. The DDPM noise scheduler handles the noise between the t's. We observe that this setup already gives a powerful boost in PSNR. **(iii) PD + UNET:** PD is added to the pipeline of (ii) acting as a denoiser, predicting $\hat{x}_0$ at time $t$. Due to the small receptive field, the network does not know global structure, effectively destroying it at large $t$'s. The UNet compensates the loss of structure induced by PD. **(iv) Kernelfusion:** Applying the (same) UNet twice, as described in Sec. 4, leads to the best results. The UNet now gets the structural information from the previous $t$, making the the optimization easier, while exploiting the patch-distribution adjusted information.

Table 3: Ablations of reverse diffusion

| Method | Blind144 PSNR↑ |
|---|---|
| DIP | 23.663 |
| UNet only | 25.804 |
| PD + UNet | 25.481 |
| KernelFusion | 27.191 |

**Real-World Images:** To evaluate the performance of our method under *realistic* degradation conditions, we utilize a variety of real-world images from various different sources: (i) The publicly available real-world benchmark of Lai et al. (2016), which consists of naturally degraded images captured in uncontrolled settings from different cameras. (ii) Historic old photos downloaded from the internet. (iii) Real blurry photos from Levin et al. (2009). (iv) Additionally, we captured real images using a high-quality DSLR camera where the *optical stabilization was disabled* on purpose. This setup introduces realistic degradations such as mild motion blur and hand tremor – common issues in everyday photography. The above collection of *diverse* real-world photos from different independent sources, taken under completely different settings, makes it ideal for testing blind-SR methods in practical scenarios. Fig. 7 shows visual comparisons of 4× SR of our method and the

*leading* state-of-the-art Blind-SR baselines. Being real images, there is no GT to compare against. Our method reconstructs more accurate details, legible text, well-defined edges, and fine textures that are heavily blurred or distorted in the competing Blind-SR methods. The complex, non-Gaussian kernels estimated by our method, highlight the importance of assumption-free blind SR. More examples are found in the appendix.

**Limitations:** While *KernelFusion* can super-resolve LR images obtained under severe *downscaling* degradations (previously considered impossible), it suffers from several limitations: (i) Operating within the standard SR formulation (Eq. 1), the very quest to estimate the downscaling kernel already implies that a unique, globally uniform kernel exists; hence KernelFusion cannot yet handle spatially varying blur. (ii) Our method relies solely on the internal statistics of a single natural image (the LR image), without leveraging pre-trained large-scale diffusion models. Although this *"assumption-free"* design is central to our ability to tackle previously intractable downscaling degradations (has no "out-of-distribution" data), it does not exploit rich external information which could further enhance its reconstruction quality. (iii) Runtime: Moreover, Patch-Diffusion (PD) has to be trained from scratch on each new LR image, currently taking $\sim$20 mins per image (on a single L40S GPU). The subsequent upscaling process depends on input image size and the scale factor applied. Exploring ways to integrate *external* learned priors with our *internal* PD model may significantly speed up KernelFusion. This represents an exciting future research direction.

## 6 CONCLUSION

"KernelFusion" recovers the unique image-specific SR-kernel directly from the LR input image, while simultaneously recovering its corresponding HR image in a consistent manner. It can handle complex downscaling degradations, where existing SotA Blind-SR methods struggle. Being a zero-shot estimation method which trains *internally* on the LR input image only, KernelFusion is not bound by any external training distribution, hence can handle any type of downscaling kernel. It has no notion of "out-of-distribution" data which *externally-trained* Blind-SR methods suffer from. By breaking free from predefined kernel assumptions, KernelFusion pushes Blind-SR into a new, unrestricted kernel paradigm – handling downscaling degradations previously thought impossible.

## 7 ACKNOWLEDGMENT

This project was funded in part by the European Union (ERC grant agreement No 101142115). A.S. is a Chaya Fellow, supported by the Chaya Career Advancement Chair.

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

# Appendix

## A  FURTHER TECHNICAL DETAILS

### A.1  ALGORITHM

---

**Algorithm 1** Reverse Diffusion Process in KernelFusion

---

1: **Input:**
2:     Pretrained velocity model: $\text{PD}_\psi$
3:     Number of noise/denoise timesteps: $T_{nd}$
4:     Noise schedule: $\{\beta_t\}_{t=1}^T$ with $\alpha_t = 1 - \beta_t$
5:     Initial guess: $\hat{x}_{0,T_{nd}} = I_{LR} \uparrow_{bic,s}$
6:     Networks: U-Net $\text{R}_\theta$, SIREN $\text{INR}_\phi$ with grid $g$
7:     Center of mass function: COM
8:     Number of optimization steps per $t$: $n_{iter}$
9:     Learning rate: $\gamma$
10: **Output:** Generated sample $x_0$
11: **for** $t = T_{nd}$ to 1 **do**
12:     $\xi \sim \mathcal{N}(0, I)$                                          → Sample noise
13:     **for** $i = 1$ to $n_{iter}$ **do**
14:         $\hat{k}_\phi \leftarrow \text{INR}_\phi(g)$                              → Get kernel
15:         $\hat{x}_{0,t+1,\theta} \leftarrow \text{R}_\theta(\hat{x}_{0,t+1})$                → Optimize $x_0$
16:         $\hat{x}_{0,t+1} \downarrow_s = (\hat{x}_{0,t+1,\theta} * \hat{k}_\phi) \downarrow_s$          → Downscale step
17:         $\hat{x}_{t,\theta} = \mu_{t+1}(\hat{x}_{0,t+1,\theta}, x_{t+1}) + \sigma_{t+1}\xi$
18:         $\hat{v}_\theta \leftarrow \text{PD}_\psi(\hat{x}_{t,\theta}, t)$                        → Denoising step
19:         $\hat{x}_{0,t,\theta} = \sqrt{\bar{\alpha}_t} \cdot \hat{x}_{t,\theta} - \sqrt{1 - \bar{\alpha}_t} \cdot \hat{v}_\theta$
20:         $\hat{x}_{0,t,\theta} \leftarrow \text{R}_\theta(\hat{x}_{0,t,\theta})$                    → Optimize $x_0$
21:         $\hat{x}_{0,t} \downarrow_s = (\hat{x}_{0,t,\theta} * \hat{k}_\phi) \downarrow_s$              → Downscale step
22:         $\mathcal{L} = \|I_{LR} - \hat{x}_{0,t} \downarrow_s \|_2^2 + \|I_{LR} - \hat{x}_{0,t+1} \downarrow_s \|_2^2 + \text{COM}(\hat{k}_\phi)$
23:         $\theta \leftarrow \theta - \gamma \nabla_\theta(\mathcal{L})$                        → Step for U-Net
24:         $\phi \leftarrow \phi - \gamma \nabla_\phi(\mathcal{L})$                        → Step for INR
25:     **end for**
26:     $\mu_t = \beta_t \frac{\sqrt{\bar{\alpha}_{t-1}}}{1-\bar{\alpha}_t}\hat{x}_{0,t,\theta} + (1 - \bar{\alpha}_{t-1})\frac{\sqrt{\alpha_t}}{1-\bar{\alpha}_t}\hat{x}_{t,\theta}$
27:     $\zeta \sim \mathcal{N}(0, I)$                                          → Sample noise
28:     $x_{t-1} = \mu_t + \sigma_t\zeta$                              → Add stochasticity, $\sigma_0 = 0$
29: **end for**
30: **return** $x_0$

---

### A.2  PATCH-DIFFUSION

**Architecture**  The backbone model is a convolutional network, that inputs an image tensor $x$ and a timestep $t$. We have a total of 6 blocks, each block conditioned on $t$. We use one block of two $3 \times 3$ filters, followed by five blocks of $3 \times 3 + 1 \times 1$ filters. We use 128 filters for the hidden layers.

**Training Details**  The patch diffusion model is trained using random crops of size 64 pixels. The model is trained for $600'000$ steps, using Adam as optimizer, with a learning rate of $lr = 1 \times 10^{-4}$ and cosine annealing. Network weights are initiated using the default configurations of Pytorch.

### A.3  U-NET

**Architecture**  The refinement UNET consists of 5 blocks, with 32 filters on the input level and 512 filters on the bottom level. Each block consists of 2 convolutional layers with a $3 \times 3$ kernel, ReLU activations and batch norm. The final layer uses a tanh activation function to ensure that the predicted $x_0$ output is in the expected -1 to 1 range. The levels are down respectively upsampled by a scale factor of 2.

**Training Details**    The U-NET is trained at each time step $t$ during the reverse diffusion process. We use Adam optimizer, a learning rate of $lr = 1 \times 10^{-4}$. We apply cosine annealing, reducing the learning rate at each $t$ to a final $lr = 1 \times 10^{-5}$. The U-NET is initialized at the first $T_{start}$ of the reverse diffusion process and then finetuned along the different timesteps $t$. With the exception of the initial $T_{start}$ where we apply $n_{iter} = 100$ iterations, the model is then finetune for $n_{iter} = 20$ iterations during each $t$. Network weights are initiated using the default configurations of Pytorch.

### A.4   Implicit Neural Representation for Kernel Estimation

**Architecture**    As described in Sec. Kernel Estimation, we took inspiration from SIREN Sitzmann et al. (2020) for our implicit neural representation. The network consists of 5 fully connected layers, with 256 nodes each. In contrast to the original paper, we reduced $\omega$ from 30 to 5 and apply it across all layers. Our last layer has an activation function which we call *leaky sigmoid*, a sigmoid function also allowing for slight negative values: $\sigma_{leaky}(x) = (1 + 10^{-4}) \cdot \sigma(x) - 10^{-4}$. The kernel is normalized such that its sum equals to 1. Additionally, as shown in Algorithm 1, a center of mass loss is introduced, encouraging the resulting kernel to be mass centered.

**Training Details**    As we train our INR along with the U-NET, we use the same training setup as described in Sec. A.3.

## B   Additional Visual Examples

### B.1   Additional Kernel Evaluation Results

Fig. 8 shows the results of the estimated kernels for 4 additional images providing additional evidence of the effectiveness of KernelFusion compared to current competitors. Fig. 10 provides a complete overview over all estimated kernels of the Blind144 dataset.

### B.2   Additional Real Life Examples

Fig. 11 to 18 show additional visual comparisons of Super-Resolution applied on a sample of diverse *real-world* images, taken from a variety of different sources (the source of each real LR image is mentioned in each example). Our recovered SR-Kernel is also provided for each image. Since these are real images, there are NO ground-truth HR images nor ground-truth kernels.

## C   Additional Ablations

### C.1   Kernel Network

As described in Sec. 4, KernelFusion leverages an Implicit Neural Representation (INR), specifically a SIREN Sitzmann et al. (2020) architecture. Alternative architectural choices are possible. In the following, two alternative choices are discussed: Directly optimizing the kernel and using a linear CNN as e.g. used in KernelGAN Bell-Kligler et al. (2019).

**Direct Kernel Optimization:**    Each entry of the $24 \times 24$ kernel grid is treated as an individual parameter that is directly optimized. Note that this means that no additional network is applied. Additionally, the optimized kernel positivity (e.g. using a ReLU Agarap (2018) activation function) and a normalization such that its sum equals to 1 is applied.

**Linear CNN:**    KernelGAN Bell-Kligler et al. (2019) leveraged a deep linear CNN for kernel estimation. This network consists solely of convolutional layers without any additional activations functions. The network is used to downscale the image and implicitly captures the kernel as such. It can be explicitly extracted by convolving all layers sequentially (see Bell-Kligler et al. (2019) for more details). Our network consists of 6 convolutional layers with kernel sizes, $[7, 7, 5, 3, 1, 4]$, channels $[1, 32, 32, 32, 32, 32, 1]$ and strides $[1, 1, 1, 1, 1, 4]$. Not that the last stride entry indicates the scale factor of 4 used in our experimental setup.

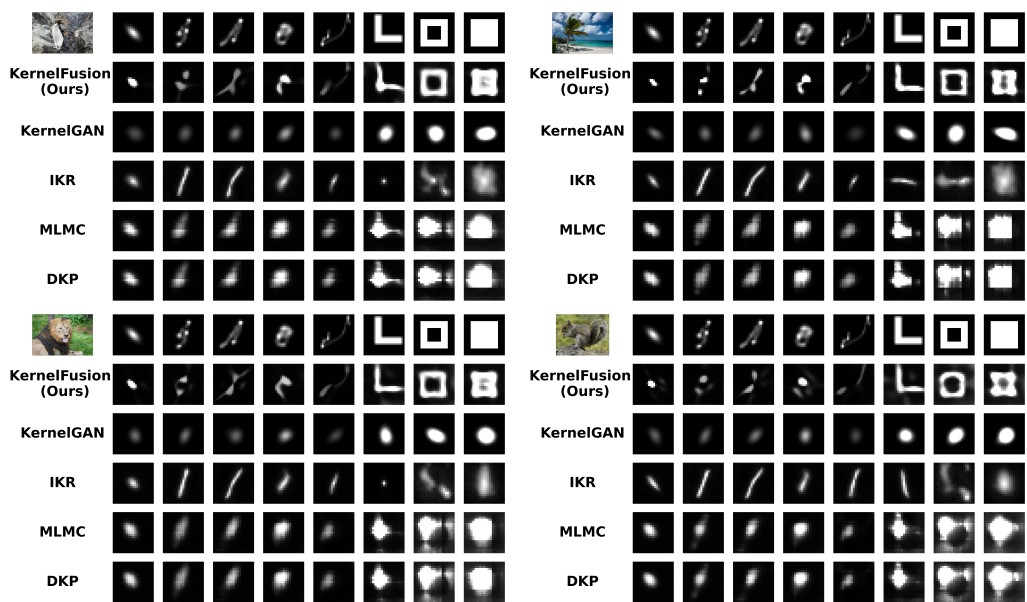

Figure 8: **Comparison of estimated kernels from different Blind-SR methods.** The top row represents the ground-truth (GT) degradation kernels, while each subsequent row corresponds to the estimated kernels from different SR methods, including our approach, KernelGAN, IKR, MLMC, and DKP. Our method demonstrates superior flexibility in recovering complex, non-Gaussian degradations, accurately capturing kernel structures across a diverse range of degradations.

Table 4: Kernel Network ablations on K0, K1, K2 of Blind144

| Method | KernelFusion PSNR↑ | DIP PSNR↑ |
|---|---|---|
| Direct Sigmoid | 16.366 | 15.529 |
| Direct Relu | 26.541 | 24.266 |
| Linear CNN | 23.494 | 23.098 |
| INR | 26.995 | 23.638 |

**Experimental Setup:** As shown in Table 4, we compare four different architectural choices for the kernel estimation: A direct kernel optimization (one with a Sigmoid and one with a ReLU activation function), a KernelGAN like linear CNN, and the SIREN INR applied in KernelFusion. To better show the power of our diffusion setup, we additionally evaluated the architectures in a DIP setup (as used in the ablations in Sec. 5).

**Results:** Table 4 shows the resulting SR PSNRs on the synthetic kernels $k_0$, $k_1$, and $k_2$ of Blind144 (see Fig. 19 and Fig. 20 for reference). KernelFusion with the INR shows superior results over the alternatives. Nevertheless, the direct optimization using a ReLU activation shows suprisingly good results. However, when comparing the reconstructed kernels (see Fig. 19 and Fig. 20) it becomes clear that the direct optimization leads to noisy, non-consistent kernels. Using an INR, much more accurate kernels can be recovered.

## C.2 Applying the U-Net twice

As discussed in Sec. 4 and in Algorithm 1, the same U-Net is applied twice, once before and once after PD. An alternative design choice could consist of two separate, identical U-Nets that do not share weights and are optimized individually with an identical parameter setup. Table 5 shows the result of the described ablation. With a PSNR of 26.498 dB, optimizing two separate U-Nets is significantly worse than optimizing the a single U-Net that has been applied twice. The intuition

Table 5: Two-UNet ablation: Weight sharing across both stages improves performance over using two independently trained U-Nets.

| Method | Blind144 PSNR↑ |
|---|---|
| KernelFusion (Two separate U-Nets) | 26.498 |
| KernelFusion (same U-Net applied twice) | 27.191 |

behind applying the same U-Net twice can be described as follows: Once the target prediction is sufficiently good at small $t$'s, the U-Net output difference between $x_0$ at $t + 1$ (before PD) and $t$ (after PD) should be marginal as the same images will have the same patch distribution (also compare Fig. 21). On the other side, we initialize Phase 2 with a bicubic guess of for $x_0$. Due to the high noise level, PD destroys global structure while adjusting the patch distribution and the second application of the U-Net receives a heavily corrupted input. During this early phase, the first application of the U-Net effectively acts as a regularizer that leverages the bicubic guess to provide the global structure information.

## D    RECEPTIVE FIELD OF PATCH DIFFUSION NETWORK

As discussed in Sec. 4 and Sec. 5, PD operates with a receptive field of $15 \times 15$ pixels. Consequently, PD learns local structure at the patch level and has no explicit notion of global image layout. When predicting $\hat{x}_0$ at large $t$ (high noise levels), this can cause the reconstruction to lose the exact global structure of the image. Fig. 21 illustrates this: for small $t$, the noised image still preserves the global structure, and PD produces a reasonable reconstruction. However, for large $t$ (e.g., $t = 800$), the input to PD is heavily corrupted, and PD reconstructs an image whose local patches match the learned LR patch distribution, but the global structure becomes strongly blurred and inconsistent, yielding a mosaic of plausible local patches. This effect diminishes as $t$ decreases and more of the original structure remains visible. Introducing a U-Net as a global image prior compensates for this lack of global awareness, allowing us to both adjust the patch distribution and recover an accurate reconstruction.

## E    RUNTIME IMPROVEMENTS

As pointed out in *Limitations* in Sec. 5, KernelFusion has to be trained from scratch for each image individually and hence comes with a training cost that is reflected in runtimes that are longer than other SOTA SR algorithms. While KernelFusion is not yet fast enough for commercial deployment, it serves a different purpose: establishing the scientific feasibility of a task previously considered impossible; Blind-SR with unrestricted kernels.

That said, optimization is a logical next step. Table 6 shows the SR PSNRs using a patch diffusion model trained for 100k steps as well as the 600k setup used for our main evaluation. With just 100k steps, patch diffusion can capture much of the necessary statistics. This yields a $4\times$ speedup ($\sim 5$ minutes compared to $\sim 20$ minutes) with negligible performance loss ($\sim 0.07$ dB).

As runtimes for KernelFusion are above commercially deployed algorithms, two points shall be additonally stressed:

**(i) Breaking Assumptions:** As shown in Fig. 1, SOTA methods rely on fixed priors (Gaussian/Motion) and collapse when these assumptions break. Our paper points out this general limitation and brings it to the attention of the community. KernelFusion succeeds in this "impossible" regime by extracting the prior directly from the LR test image. We note that other single-image / internal-learning methods (e.g., KernelGAN Bell-Kligler et al. (2019), MLMC Xia et al. (2024), DKP Yang et al. (2024b)) also require per-image optimization and typically take several minutes per image to produce their SR output. Yet, none of these can handle arbitrary downscaling kernels.

**(ii) Feasibility:** We see a parallel to early foundational works like NeRF Mildenhall et al. (2021) or Deep Image Prior Ulyanov et al. (2018), which required significant computation to establish feasibility in new domains. By proving that unrestricted kernel recovery is solvable, KernelFusion

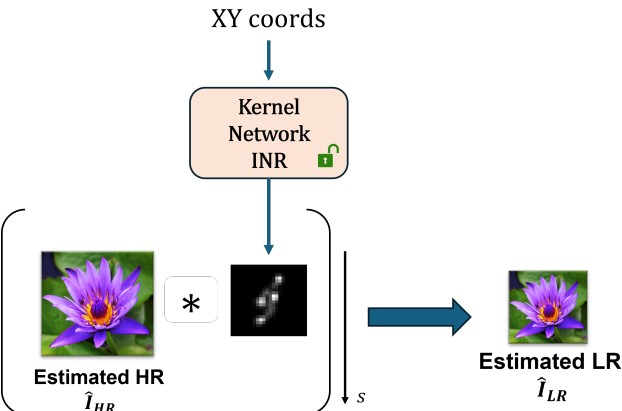

Figure 9: **Illustration of the kernel estimation in Phase 2.** The INR-based kernel network maps normalized $(x, y)$ coordinates to kernel values, producing a downscaling kernel that is convolved with the current HR estimate and downsampled by a factor $s$ to yield the predicted LR image $\hat{I}_{\text{LR}}$.

Table 6: Runtime improvements: Training PD for 100k steps instead of 600k steps leads to marginal losses in PSNR.

| Method | Blind144 PSNR↑ | Runtime min↓ |
|---|---|---|
| KernelFusion (PD: 100K) | 27.117 | $\sim 5$ |
| KernelFusion (PD: 600K) | 27.191 | $\sim 20$ |

opens a new, free from predefined kernel assumptions paradigm. We hope that the community will build on these foundations to bridge the gap to real-time efficiency.

## F  DETAILS ON KERNEL ESTIMATION IN PHASE 2.

Given the current HR estimate $\hat{I}_{\text{HR}}$ and the INR-based kernel network $\text{INR}_\theta$, we represent the downscaling kernel on a $K \times K$ grid (in our experiments $K = 24$). For each normalized kernel coordinate $g$ we evaluate

$$k_\theta = \text{INR}_\theta(g), \tag{3}$$

To obtain a valid kernel, we enforce non-negativity and unit sum. Using this kernel, the predicted LR image $\hat{I}_{\text{LR}}$ is obtained by convolving the current HR estimate with $\hat{k}_\theta$ and then downsample by the scale factor $s$ (here $s = 4$):

$$\hat{I}_{\text{LR}} = (\hat{I}_{\text{HR}} * \hat{k}_\theta) \downarrow_s, \tag{4}$$

where $\downarrow_s$ denotes subsampling by $s$ and $*$ represents a discrete convolution.

The patch-diffusion model in Phase 1 was trained on the observed LR image $I_{\text{LR}}$. Therefore, in Phase 2 we enforce consistency between the predicted LR image and the original observed LR image via an MSE loss,

$$\mathcal{L}_{\text{cons}} = \left\| \hat{I}_{\text{LR}} - I_{\text{LR}} \right\|_2^2 \tag{5}$$

which couples the HR estimate $\hat{I}_{\text{HR}}$ and the kernel parameters $\theta$ through the forward model. Figure 9 illustrates the kernel estimation process.

## G  USE OF LARGE LANGUAGE MODELS

Large Language Models (LLMs) have been used while writing the paper i.e. to identify typos, improve formulations or shorten text snippets.

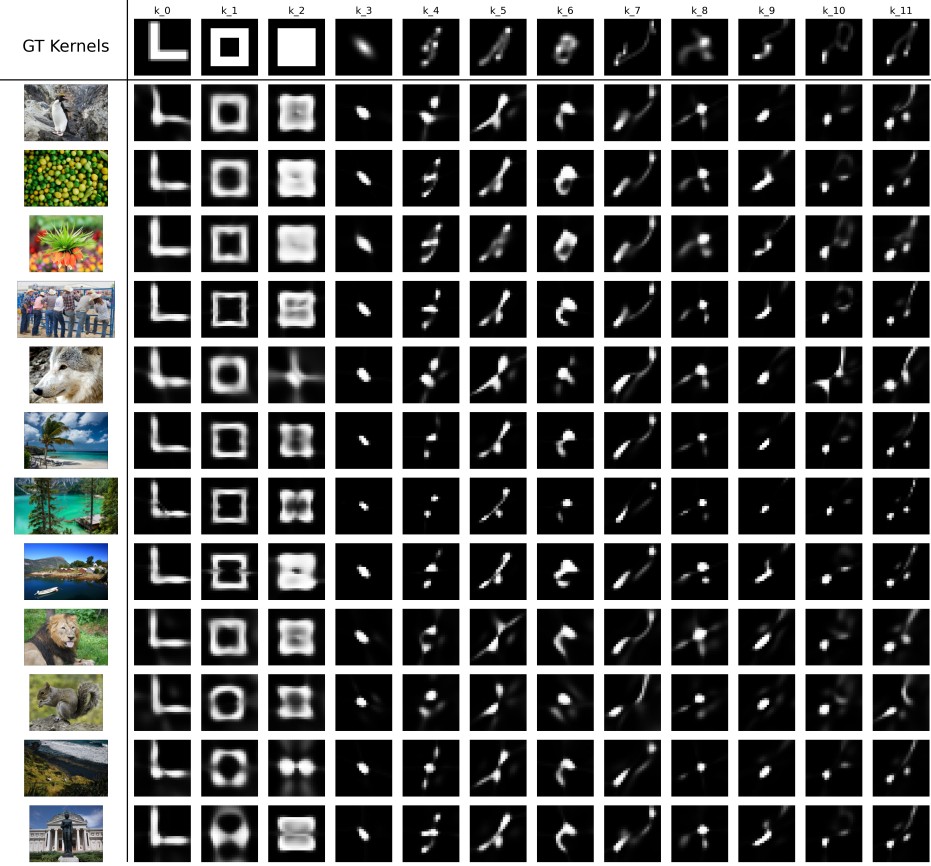

Figure 10: **Estimated kernels of KernelFusion on Blind144:** Complete overview of all 144 estimated kernels.

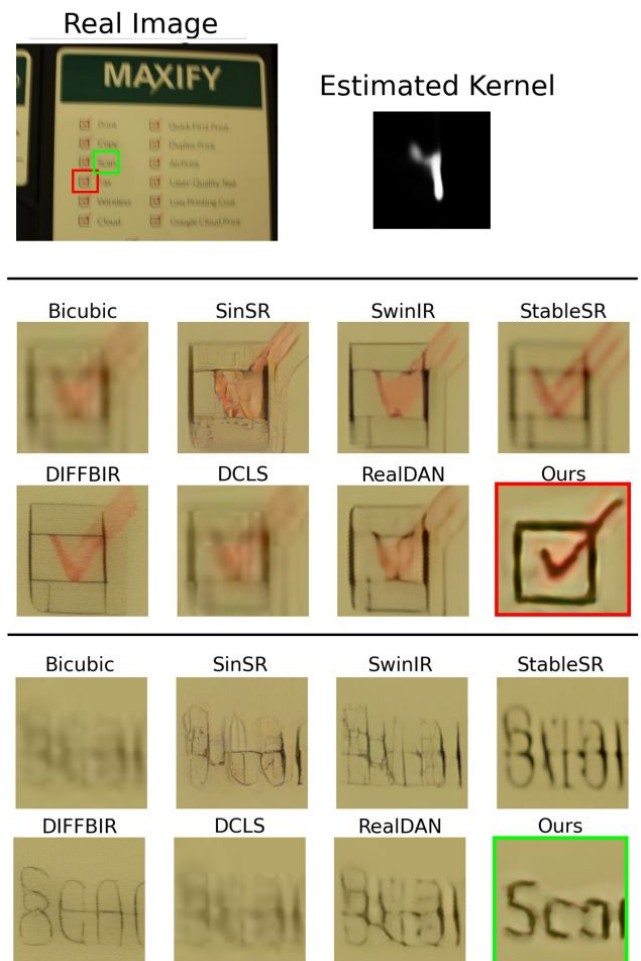

Figure 11: **Super-Resolution of a REAL image (taken from Lai et al. (2016)).**

(*Top*)   **The real image (used as the LR input) & its estimated SR-kernel (using our method).**

(*Middle & Bottom*)   **A visual comparison of SR×4 results of leading SR methods. Zooms of the red and green image regions are displayed, respectively.** *(There is NO ground-truth HR image)*

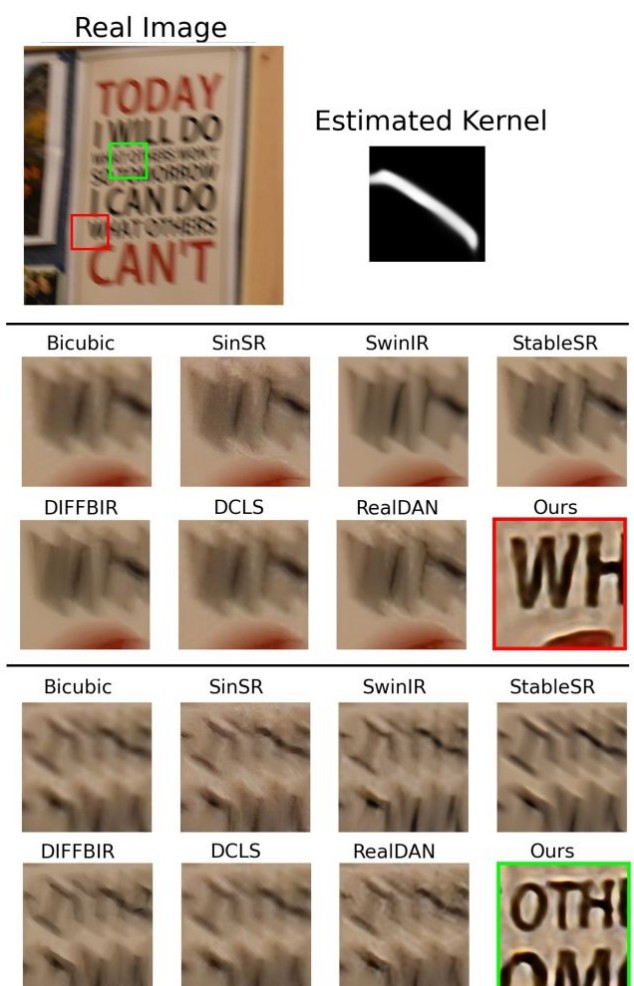

Figure 12: **SR of a REAL image (captured using a DSLR camera *without* optical stabilization).**

(*Top*)  **The real image (used as the LR input) & its estimated SR-kernel (using our method).**

(*Middle & Bottom*)  **A visual comparison of SR×4 results of leading SR methods. Zooms of the red and green image regions are displayed, respectively.**  *(There is NO ground-truth HR image)*

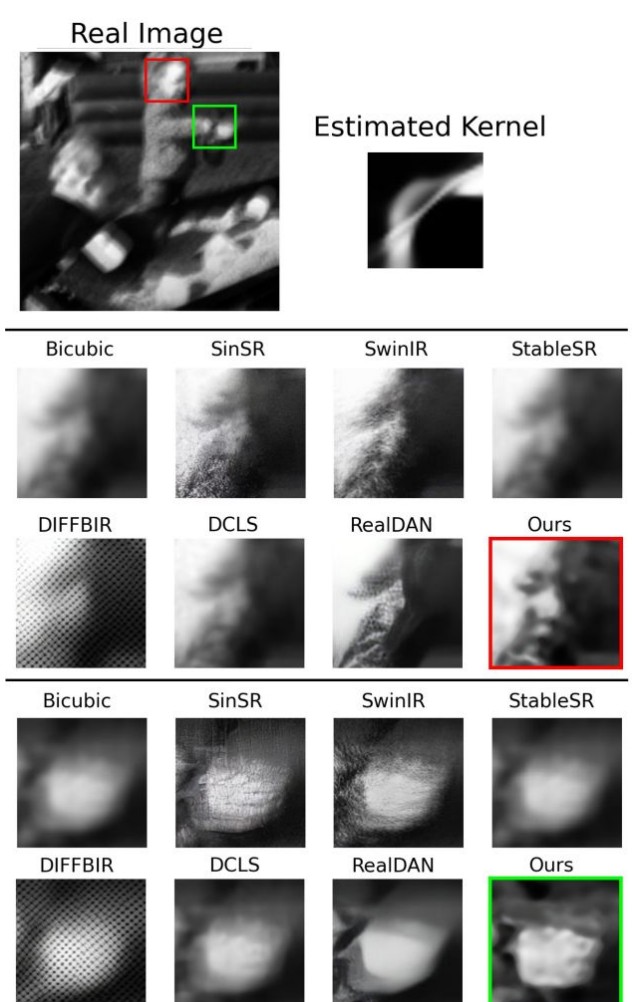

Figure 13: **Super-Resolution of a REAL image (taken from Levin et al. (2009)).**

(*Top*)  **The real image (used as the LR input) & its estimated SR-kernel (using our method).**

(*Middle & Bottom*)  **A visual comparison of SR×4 results of leading SR methods. Zooms of the red and green image regions are displayed, respectively.**  *(There is NO ground-truth HR image)*

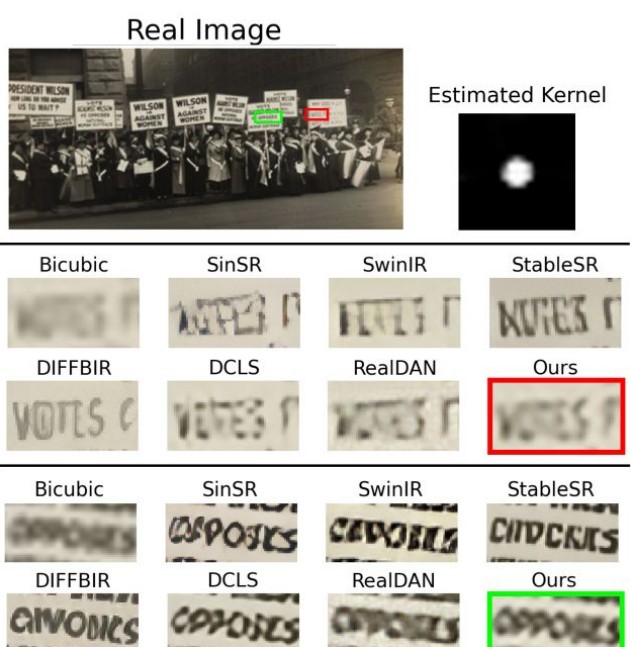

Figure 14: **SR of a REAL old (poor-quality) historic photo downloaded from the internet.**

(*Top*)   **The real image (used as the LR input) & its estimated SR-kernel (using our method).**

(*Middle & Bottom*)   **A visual comparison of SR×4 results of leading SR methods. Zooms of the red and green image regions are displayed, respectively.**  *(There is NO ground-truth HR image)*

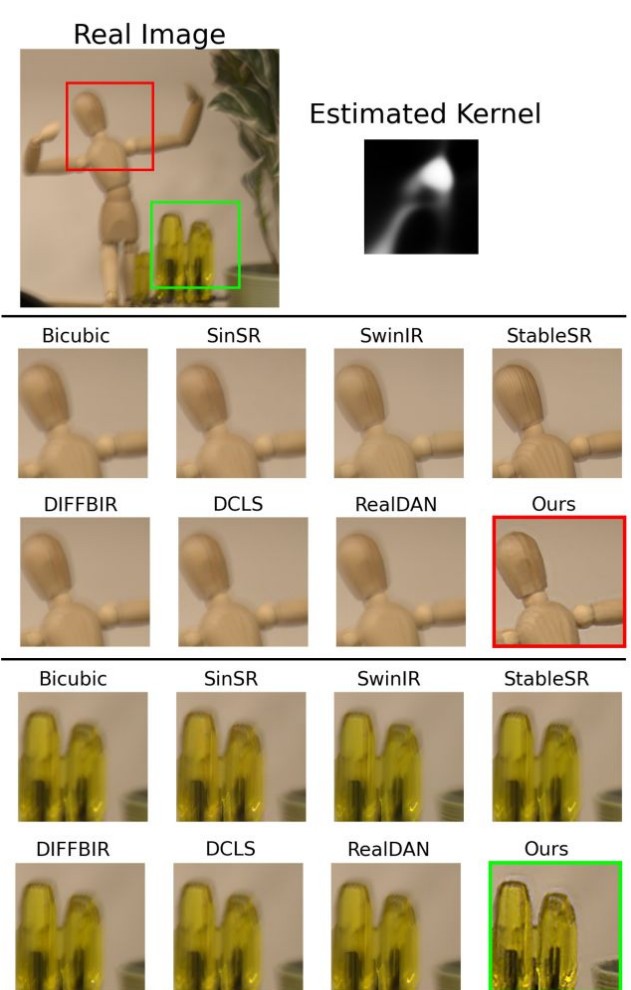

Figure 15: **SR of a REAL image (captured using a DSLR camera *without* optical stabilization).**

(*Top*)  **The real image (used as the LR input) & its estimated SR-kernel (using our method).**

(*Middle & Bottom*)  **A visual comparison of SR×4 results of leading SR methods. Zooms of the red and green image regions are displayed, respectively.**  *(There is NO ground-truth HR image)*

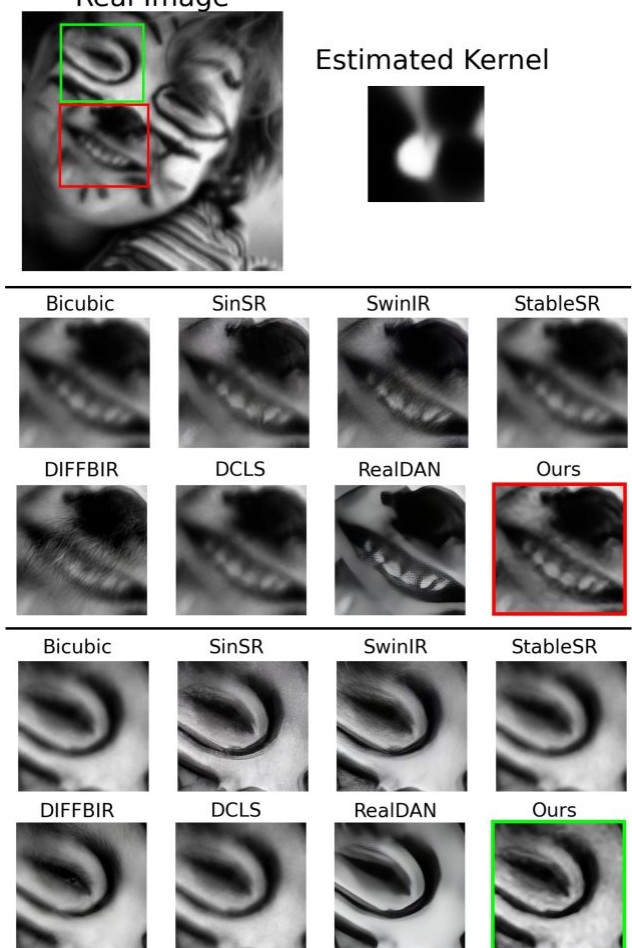

Figure 16: **Super-Resolution of a REAL image (taken from Levin et al. (2009)).**

(*Top*) **The real image (used as the LR input) & its estimated SR-kernel (using our method).**

(*Middle & Bottom*) **A visual comparison of SR×4 results of leading SR methods. Zooms of the red and green image regions are displayed, respectively.** *(There is NO ground-truth HR image)*

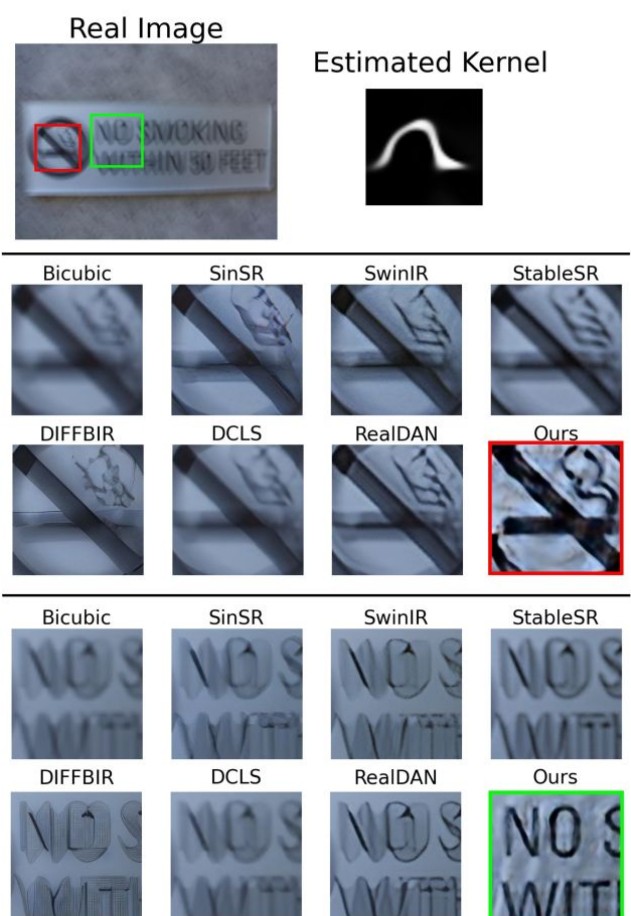

Figure 17: **Super-Resolution of a REAL image (taken from Lai et al. (2016)).**

(*Top*)  **The real image (used as the LR input) & its estimated SR-kernel (using our method).**

(*Middle & Bottom*)  **A visual comparison of SR×4 results of leading SR methods. Zooms of the red and green image regions are displayed, respectively.** *(There is NO ground-truth HR image)*

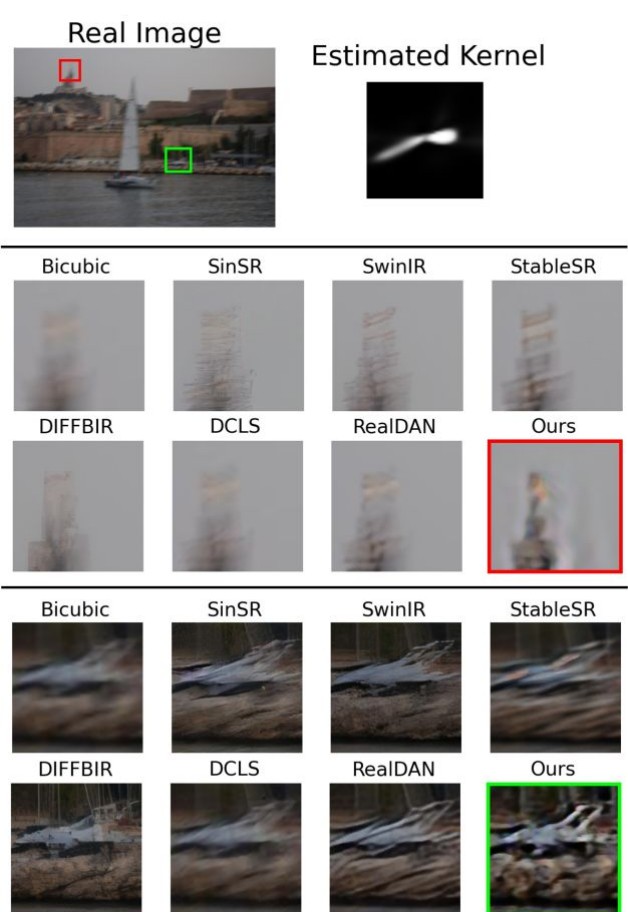

Figure 18: **Super-Resolution of a REAL image (taken from Lai et al. (2016)).**

(*Top*)  **The real image (used as the LR input) & its estimated SR-kernel (using our method).**

(*Middle & Bottom*)  **A visual comparison of SR×4 results of leading SR methods. Zooms of the red and green image regions are displayed, respectively.** *(There is NO ground-truth HR image)*

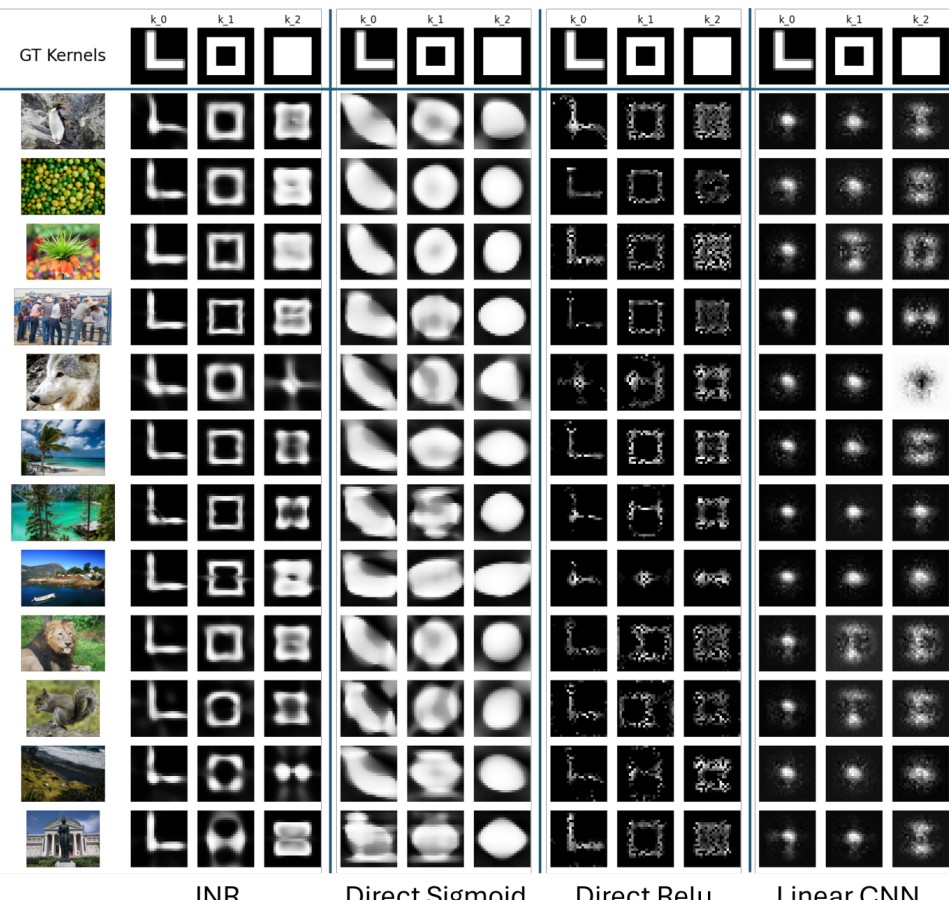

Figure 19: **Estimated kernels for KernelFusion setup with 4 different kernel estimation approaches:** INR, direct kernel estimation with sigmoid activation, direct kernel estimation with ReLU activation, and linear CNN inspired by KernelGAN.

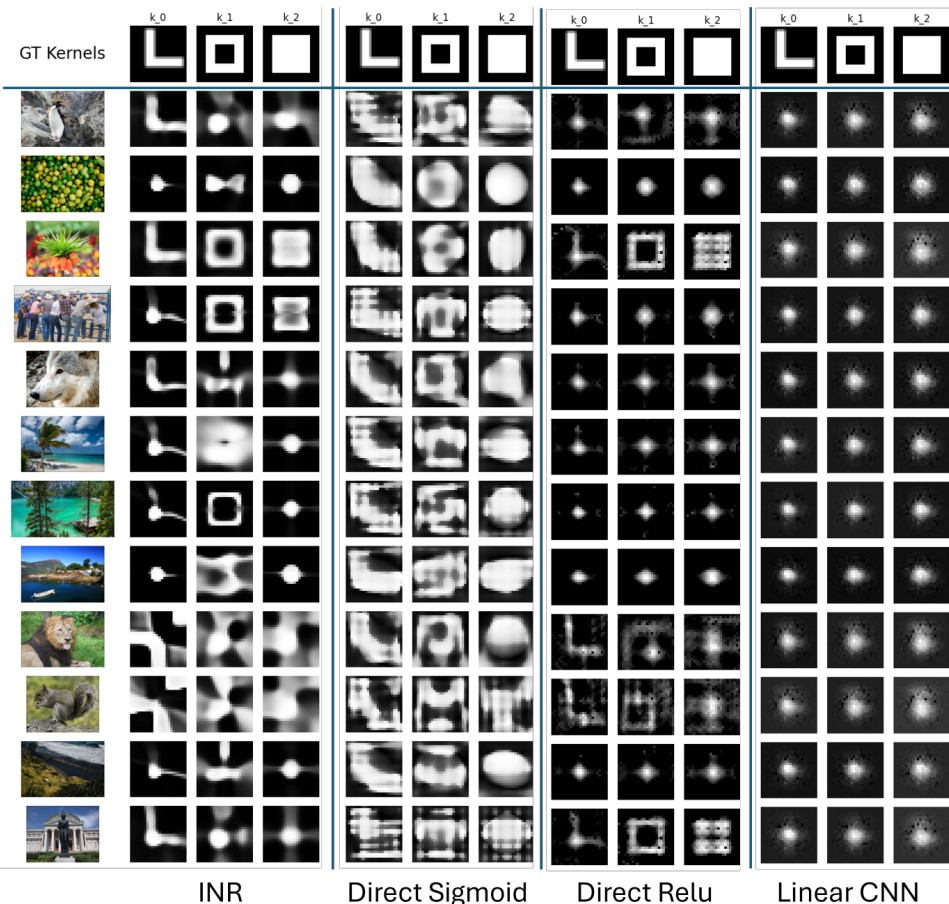

Figure 20: **Estimated kernels for the DIP setup with 4 different kernel estimation approaches:** INR, direct kernel estimation with sigmoid activation, direct kernel estimation with ReLU activation, and linear CNN inspired by KernelGAN.

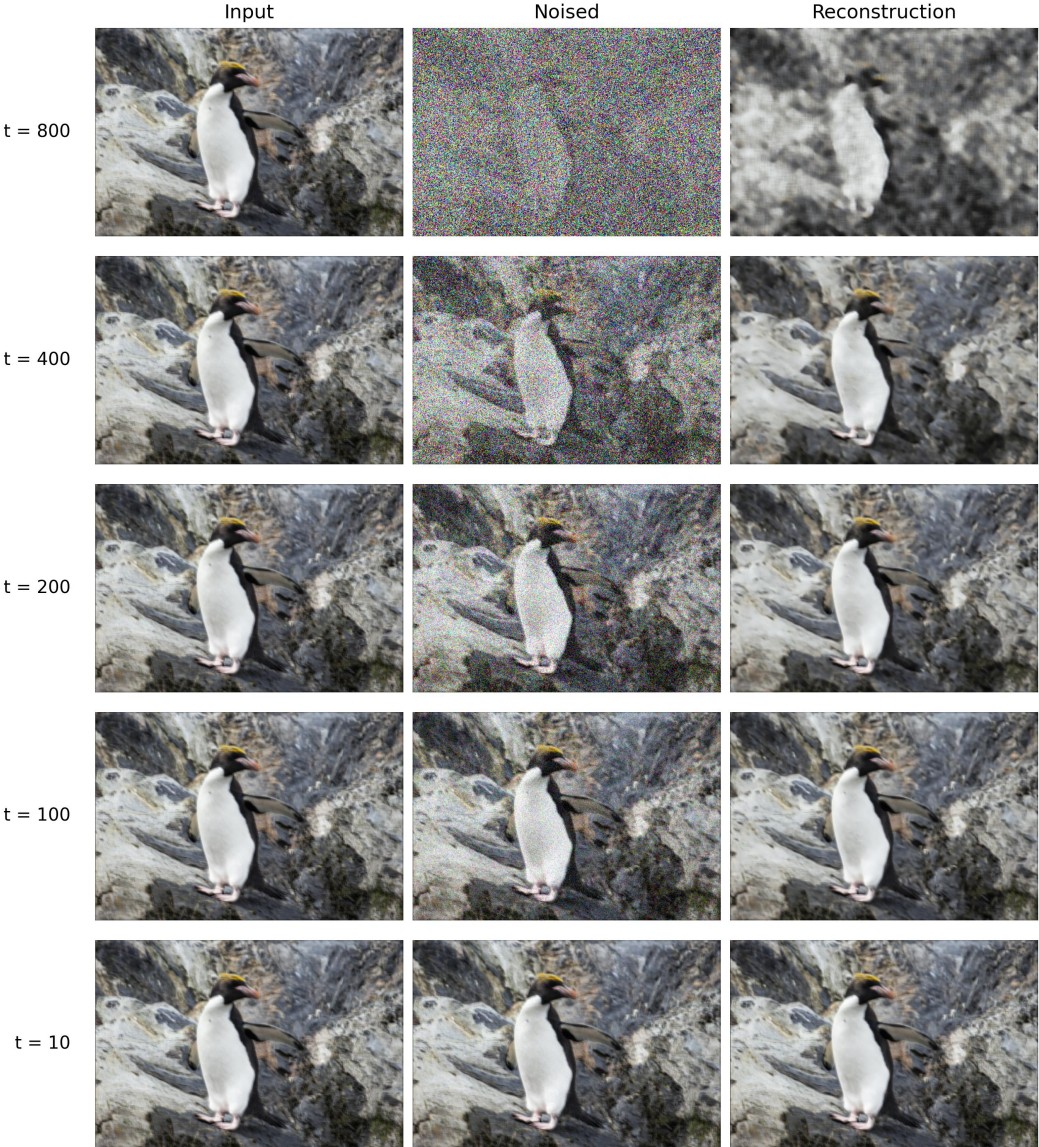

Figure 21: **Prediction of $\hat{x}_0$ using Patch Diffusion:** At high noise levels - corresponding to large $t$'s - patch diffusion struggles to maintain global structure, and produces a mosaic-like reconstruction of local patches (best viewed zoomed in).

