# OpenReview forum: "KernelFusion: Zero-Shot Blind Super-Resolution via Patch Diffusion"
_ICLR.cc/2026/Conference — ICLR 2026 Poster_

### Official Review · Reviewer_t2tb · 2025-10-26

**Soundness:** 3
**Presentation:** 1
**Contribution:** 2
**Rating:** 4
**Confidence:** 4

**Summary:**

This paper introduces a zero-shot blind image super-resolution method that leverages a diffusion model to learn patch distribution and then further use it for kernel estimation and super-resolution. The overall approach contains two stages: patch diffusion training and blind super-resolution. In the second stage, an image-specific kernel is estimated with implicit kernel representation. In experiments, the proposed method is evaluated on three blind SR datasets with complex kernels.

**Strengths:**

- The proposed zero-shot training and inference is interesting, and the predicted kernels look good.
- The results on Blind144 and DIV2KFK is promising.

**Weaknesses:**

Main weaknesses:
- In Figure 2, the HR images are sent to the UNet twice at each iteration. The training and inference of diffusion model s also makes this algorithm inefficient. Could the authors provide a comparison on model complex and inference time?
- Sec. 4 is not very clear to me, particular for the Phase 2. Could you rephrase it and provide a figure (or some math equations) for the kernel estimation part?
- The experiments are only performed on x4 SR, it would be better to also evaluate the method on x2 or x8 SR datasets.
- The written can be improved.
- Small issue and suggestion: please use "\citep" rather than "\cite" for the citations.

**Questions:**

- The diffusion is trained with T=1000 steps, have you tried to use some acceleration algorithms such as DDIM to reduce the inference time?
- In Eq. (1), the degradation is design without noise, but what if the LR image contains small noise?
- How do you design the kernel size you want to learn in real world applications?

---

> ### Author Response · Authors · 2025-11-21
>
> We thank the reviewer for the time and feedback. We addressed the raised weaknesses (W) and questions (Q), supported by additional experimental data.
>
> ***
> **W1. Efficiency:**
> > *In Figure 2, the HR images are sent to the UNet twice at each iteration. The training and inference of diffusion model s also makes this algorithm inefficient. Could the authors provide a comparison on model complex and inference time?*
>
> We acknowledge (and also mention this in the limitations Section 5 of the submitted paper) that runtime is a limitation of our method. However, we would like to emphasize that the architectural choices are not arbitrary. We would like to point to the ablations in Section 5 and Table 3 of the submitted paper and further new ablations in the new Section C.2 of the Appendix.
>
> KernelFusion is not yet fast enough for commercial deployment, but it serves a different purpose: establishing the scientific feasibility of a task previously considered impossible; Blind-SR with unrestricted kernels.
>
> That said, optimization is the logical next step. Inspired by your feedback, we found that reducing Patch-Diffusion training to just 100k steps captures the necessary statistics. This yields a **4$\times$ speedup** (~5 mins instead of 20 mins) with negligible performance loss ($\sim0.07$ dB). We now added these results in Sec. E / Table 6 of the revised manuscript, to explicitly illustrate this cost-performance trade off and to show that the current 20 minute training per image setting is not strictly required for strong performance.
>
> Having said that, we want to stress two points:
>
> **(i) Breaking Assumptions:** As shown in Fig. 1, SOTA methods rely on fixed priors (Gaussian/Motion) and collapse when these assumptions break. Our paper points out this general limitation and brings it to the attention of the community. We further show that KernelFusion succeeds in this "impossible" regime by extracting the prior directly from the LR test image. We note that other single-image / internal-learning methods  (e.g., KernelGAN, MLMC, DKP) also require per-image optimization and typically take several minutes per image to produce their SR output. Yet, none of these can handle arbitrary downscaling kernels (as shown in Fig. 4 and 8).
> \
> **(ii) Feasibility:** We see a parallel to early foundational works like NeRF or Deep Image Prior, which required significant computation to establish feasibility in new domains. By proving that unrestricted kernel recovery is solvable, KernelFusion opens a new, assumption-free paradigm. We hope that the community will build on these foundations to bridge the gap to real-time efficiency.
>
>
> ***
> **W2. Inclarities in Sec. 4:**
> > *Sec. 4 is not very clear to me, particular for the Phase 2. Could you rephrase it and provide a figure (or some math equations) for the kernel estimation part?*
>
> We acknowledge the reviewers concern and have worked on further clarifications:
> - We have revised the Phase 2 paragraph of Section 4: We added further clarifications, explicit references to the components in Figure 2 and references to additional clarifications in the Appendix
> - We added a new Section F to the Appendix of the revised version giving a detailed explanation of the kernel estimation including detailed mathematical equations.
> - In addition to the overview Figure 2 of the submitted paper, we added a new Figure 9 to the Appendix of the revised paper illustrating explanations of Section F in more detail.
>
>
> ***
> **W3. Evaluation on x2 and x8 Scales:**
> > *The experiments are only performed on x4 SR, it would be better to also evaluate the method on x2 or x8 SR datasets.*
>
> Our goal was to ensure a fair comparison under identical conditions between the different baselines at sufficient complexities. In practice, x4 is the only scale factor shared by all competing methods. All supervised baselines (pretrained models) that we compare against in Table 2 provide weights for x4 SR. In addition, the single-image algorithms, while in principle not restricted to a specific scale, already require substantial resources at x4 (e.g. we had to run MLMC (Xie et al. 2024) on an H100 with 96GB GPU memory). For these reasons, we focused our evaluation on the common x4 scale.
>
> ***
> **W4. “The written can be improved”**
>
> Based on the reviewers input, we worked on clarifying key parts of the paper, such as Phase 2 in Section 4. We further refer to the new sections C to F in the Appendix giving additional explanations and complementary experimental data. We are open to further improve the paper and ask the reviewer to point us to sections they think are not written well.

---

> > ### Author Response · Authors · 2025-11-21
> >
> > **Question 1:**
> >
> > > *The diffusion is trained with T=1000 steps, have you tried to use some acceleration algorithms such as DDIM to reduce the inference time?*
> >
> > We appreciate the suggestion to use acceleration schemes such as DDIM to reduce inference time. While we did not explicitly implement a DDIM version, we did experiment with using a diffusion model with fewer time steps. Specifically, we trained a model using T=100 (instead of T=1000) and started the reverse diffusion loop at t=40 (instead of T=400), leaving the remaining optimization setup identical. We observed a non-negligible degradation in the SR reconstruction quality and less stable kernel estimations. On Blind144, we get the following results:
> >
> > - KernelFusion (T=1000, t_start = 400): PSNR 27.191 dB
> > - KernelFusion (T=100, t_start = 40): PSNR 24.198 dB
> >
> > Naively reducing the time steps T is not enough. For this reason, we kept the standard T=1000 schedule in the current work. That said, we agree that combining KernelFusion with DDIM style samplers could be a promising direction for future work - aimed specifically at improving efficiency.
> >
> >
> > ***
> > **Q2. Appearance of Noise in the LR Images:**
> > > *In Eq. (1), the degradation is design without noise, but what if the LR image contains small noise?*
> >
> > Eq. (1) follows the standard blind SR formulation, where the LR image is modelled as a noise free blurred and subsampled version of the HR image. In real scenarios, however, LR images contain sensor noise and other acquisition artifacts. This setting is reflected in our real-image experiments (see Fig. 7 and additional examples in the Appendix of the submitted paper), where the LR inputs do contain noise. We observe that KernelFusion is reasonably robust - it recovers sharp structures and legible text while competing methods often fail. In principle, our formulation can be extended to include an explicit noise term (e.g., additive Gaussian noise with learnable variance).
> >
> > ***
> > **Q3. Kernel Size in Real-World Applications:**
> > > *How do you design the kernel size you want to learn in real world applications?*
> >
> > In practice, the kernel size is chosen as an upper bound on the expected spatial extent of the blur in pixels in the HR image. The main design principle is that the kernel window should be large enough to fully contain the degradation. An undersized kernel would force the method to explain the LR image with an incorrect, truncated kernel. The upper bound depends on factors such as the image resolution in pixels and the expected motion magnitude influenced by e.g., handheld camera shake and exposure time. For typical scenarios a kernel size of 24x24 pixels should be sufficient. For big motion or blur, the kernel would need to be chosen larger.

---

> > > ### Comment · Reviewer_t2tb · 2025-11-24
> > >
> > > I appreciate the author's efforts in the rebuttal. Some of my questions, such as the noisy input and kernel size setup, are well addressed. However, I'm still concerned by the efficiency, clarity of method, and the lack of multi-scale SR experiments, thus I maintain my original score.
> > >
> > > In addition, since the method involves a patch-diffusion for HR estimation, I would further suggest the authors to add some perceptual metrics such as LPIPS or FID to show the ability to generate photo-realistic images in the next revision.

---

> > > > ### Author Response · Authors · 2025-12-02
> > > >
> > > > We thank the reviewer for their continued engagement and for acknowledging that our responses regarding noisy inputs and kernel sizes were satisfactory.
> > > >
> > > > Regarding the remaining concerns:
> > > >
> > > > 1. **Efficiency:** First we point out that we demonstrated a 4x speedup for PD in our new ablation (Table 6) (down to ~5 minutes) with negligible drop in PSNR. Second, we would like to reiterate that our main goal is to **establish feasibility** of blind SR with unrestricted kernels, similar in spirit to early NeRF/DIP works, rather than to optimize runtime for commercial deployment.
> > > > 2. **Multi-scale Evaluation:** We focused on x4 because we follow the **standard evaluation** protocol established in the recent Blind-SR literature. This is the mainstream setting used for fair benchmarking, as most supervised baselines (e.g., SwinIR, StableSR) provide pre-trained weights specifically for x4. Furthermore, unsupervised methods (e.g., MLMC) encounter severe hardware limitations at other scales. Therefore, x4 is the only viable intersection for a head-to-head comparison against the state-of-the-art.
> > > > 3. **Perceptual Metrics:** Our method aims to solve the inverse problem by recovering the specific degradation kernel. Therefore, our primary goal is **signal fidelity** (accurate reconstruction of the ground truth) rather than perceptual hallucination. Distortion metrics (PSNR/SSIM) are the standard proxy for measuring kernel recovery accuracy . Perceptual metrics often favor generative hallucination, which can look "realistic" even when the underlying kernel and content are incorrect, contradicting the goal of faithful restoration.
> > > >
> > > > We hope these revisions demonstrate the feasibility of the approach, which is the primary goal of this work.

---

### Official Review · Reviewer_cUZR · 2025-10-30

**Soundness:** 1
**Presentation:** 2
**Contribution:** 2
**Rating:** 2
**Confidence:** 5

**Summary:**

This paper lacks sufficient motivation and fair evaluations. The proposed method tends to resolve complex kernel, however, the simulations are carried on unfair settings. The complex kernels are not real in application.

**Strengths:**

The paper is well-written and organised in a good structure.

**Weaknesses:**

1 This paper focuses on synthetic kernels (e.g., L-shape, empty squares in Fig. 5) with no evidence of their occurrence in real-world data. Why should the community care about reconstructing images degraded by a square kernel? The authors must justify whether such kernels exist beyond synthetic stress tests.
2 Training a diffusion model for ​20 minutes per image​ on an L40S GPU is impractical for real applications (e.g., mobile or large-scale processing). The authors do not discuss trade-offs between cost and performance gains.
​3  The authors claim diffusion models excel at capturing patch statistics but fail to compare against simpler internal-learning methods (e.g., GANs in KernelGAN). What advantages does diffusion offer over GANs for this task? The justification is weak.

4 This paper omits critical details: How are the patch diffusion and INR networks initialized? What is the role of the "consistency loss" in Eq. 2? How does it interact with the diffusion prior? Why is a U-Net applied twice in Phase 2 (Sec. 4)? The ablation (Table 3) hints at its importance but lacks explanation. Kernel Estimation Stability: The INR-based kernel estimator is praised for flexibility but not evaluated for stability (e.g., sensitivity to noise or image content).

5 The authors highlight performance on non-Gaussian kernels but ​ignore standard scenarios​ (e.g., isotropic Gaussian, motion blur). Does KernelFusion underperform on common kernels? Table 2 shows it is "comparable" on DIV2KRK (Gaussian kernels) but lacks head-to-head comparisons with SOTA methods like SwinIR or Real-ESRGAN in these settings.
​6 On DIV2KFK and Blind144, KernelFusion surpasses SOTA methods by only ​0.1-0.3 dB in PSNR (Table 2), while bicubic interpolation outperforms many blind SR methods. Are these gains meaningful given the high computational cost?
​7 Fig. 7 shows real-image results but without ground truth, making it impossible to assess accuracy. The kernels estimated for real images (e.g., DSLR without stabilization) appear unstructured—are they physically plausible?

**Questions:**

See weaknesses

---

> ### Author Response · Authors · 2025-11-21
>
> We thank the reviewer for the time and feedback. Below we address the reviewer’s question and concerns.
>
> ***
>
> **W1. Synthetic Stress-Test Kernels vs. Real-World Relevance:**
>
> While the “square” kernel is a stress test, please note that our evaluation is not limited to synthetic kernels. We also use empirically measured **real motion kernels** extracted by Levin et al. (2009), which were incorporated in the DIV2KFK and Blind144 datasets, and show that KernelFusion excels at recovering these real kernels. In addition, our real-image experiments (Fig. 7 and Appendix) demonstrate that highly non-Gaussian and irregular kernels occur in real images. KernelFusion enables to handle such (real) low-quality images, recover legible text and fine structures that competing methods fail to reconstruct. This indicates that complex, unstructured kernels do occur in practice and that the ability to handle such cases is valuable for blind SR.
>
> ***
> **W2. runtime and cost–performance trade-offs:**
>
> This is a valid concern. KernelFusion is not yet fast enough for commercial deployment, but it serves a different purpose: establishing the scientific feasibility of a task previously considered impossible; Blind-SR with unrestricted kernels.
>
> That said, optimization is the logical next step. Inspired by your feedback, we found that reducing Patch-Diffusion training to just 100k steps captures the necessary statistics. This yields a $4\times$ speedup (~5 minutes instead of 20 mins) with negligible performance loss ($\sim0.07$ dB). We now added these results in Sec. E / Table 6 of the revised manuscript, to explicitly illustrate this cost-performance trade off and to show that the current 20 minute training per image setting is not strictly required for strong performance.
> Having said that, we want to stress two points:
>
>
> **(i) Breaking Assumptions:** As shown in Fig. 1, SOTA methods rely on fixed priors (Gaussian/Motion) and collapse when these assumptions break. Our paper points out this general limitation and brings it to the attention of the community. We further show that KernelFusion succeeds in this "impossible" regime by extracting the prior directly from the LR test image. We note that other single-image / internal-learning methods  (e.g., KernelGAN, MLMC, DKP) also require per-image optimization and typically take several minutes per image to produce their SR output. Yet, none of these can handle arbitrary downscaling kernels (as shown in Fig. 4 and 8).
> \
> **(ii) Feasibility:** We see a parallel to early foundational works like NeRF or Deep Image Prior, which required significant computation to establish feasibility in new domains. By proving that unrestricted kernel recovery is solvable, KernelFusion opens a new, assumption-free paradigm. We hope that the community will build on these foundations to bridge the gap to real-time efficiency.
>
> ***
> **W3. Diffusion vs. GAN-Based Internal Learning:**
>
> We would like to raise the reviewers attention that we do in fact compare against several internal-learning baselines, including the KernelGAN (a GAN-based internal method), as well as against MLMC, IKR, and DKP (non-GAN based internal learning methods). KernelFusion surpasses all of them on all three datasets, as shown in Table 2 in the submitted paper.
> More generally, diffusion models have been shown to excel at learning the internal patch distribution of a single image (e.g., SinFusion), providing a strong single image generative prior. In contrast, GAN-based internal methods such as KernelGAN are prone to mode collapse and unstable to train, especially in the single image regime. This makes diffusion models a more reliable choice for simultaneously reconstruction of the downscaling kernel and the HR image.

---

> > ### Author Response · Authors · 2025-11-21
> >
> > ***
> > **W4. Missing Implementation Details:**
> >
> > The majority of the implementation details pointed by the reviewer as missing, are actually found in the submitted paper (and are referred to below). The remaining missing details are now explained below (and are added to the revised paper):
> >
> > - Network initialization: We leverage the default weight initialization of pytorch for the different layer types. Following the reviewer's question, we now added this explanation to the Technical Details in Appendix A.  Generally, we plan to release our code upon acceptance to allow full reproducibility.
> > - Consistency loss: This is explained in the “Consistency Loss” paragraph in Sec. 4 in the submitted paper. This is the loss that ensures pixelwise alignment of the LR input and LR prediction. Please also see Algorithm 1 in the appendix.
> > - “How does it interact with the diffusion prior”: The diffusion prior (PD - PatchDiffusion) is frozen during phase 2, but gradients are propagated through the prior. We now added a clarifying footnote in the revised manuscript (new Footnote 2 on Page 6) .
> > - “Why is the U-Net applied twice”: This was shown in the paragraph “Ablations” and Table 3, in the submitted paper. Applying the UNet only once (case (iii), “PD + UNet” in Table 3) is not sufficient to compensate for the degradations introduced by PD. By applying the UNet twice, the UNet can leverage the structural information from the previous timestep and gets a significant performance boost.
> > - “INR stability”: In real scenarios, LR real images contain sensor noise and other acquisition artifacts. We explicitly test this setting in our real-image experiments (Fig. 7 in the submitted paper and additional examples in the appendix), where the LR inputs do contain noise, and observe that KernelFusion is reasonably robust: it recovers sharp structures and legible text while competing methods often fail.
> >
> > ***
> > **W5. “Standard Scenarios” and Comparison to SwinIR / Real-ESRGAN:**
> >
> > We do not ignore standard Gaussian scenarios. The DIV2KRK benchmark used in Table 2 is a commonly used dataset with pairs of LR/HR images generated using random isotropic and anisotropic Gaussian kernels. On this dataset we did report head-to-head comparisons against a broad range of blind SR baselines, **including SwinIR**. As shown in Table 2, KernelFusion is competitive on DIV2KRK compared to strongest baselines, and is **much better than SwinIR, even on Gaussian Kernels (by ~1.6dB)**. Moreover, it clearly surpasses all methods on the two non-Gaussian datasets (Blind144 and DIV2KFK). This indicated that our method handles arbitrary kernels without sacrificing performance on common Gaussian ones.
> >
> > The reviewer is correct that Real-ESRGAN is missing from Table 2. For completeness, we now also evaluated Real-ESRGAN [Wang et al. 2021] on these 3 benchmarks, obtaining:
> >
> > - DIV2KRK: PSNR 24.330, SSIM 0.678
> > - DIV2KFK: PSNR 23.003, SSIM 0.617
> > - Blind144: PSNR 23.646, SSIM 0.610
> >
> > Thus, **Real-ESRGAN performs substantially worse than KernelFusion (by several dBs!) on all 3 datasets**, and in fact even below bicubic interpolation on the challenging non-Gaussian ones. Once going out of Real-ESRGAN training distribution, its performance degrades. In contrast, KernelFusion does not rely on any external training data and is optimized per image, so there is no train/test distribution shift by construction. The revised paper now includes also the results on Real-ESRGAN in Table 2.
> >
> > ***
> > **W6. PSNR Gains**:
> >
> > > *On DIV2KFK and Blind144, KernelFusion surpasses SOTA methods by only ​0.1-0.3 dB in PSNR (Table 2).*
> >
> > **Factual Correction:** We respectfully point out a factual error in the reviewer’s assessment of Table 2. On the two non-Gaussian benchmarks (Blind144 and DIV2KFK), KernelFusion improves over both bicubic interpolation and prior blind SR methods **by more than 2 dB** (not 0.1–0.3 dB). On challenging blind SR benchmarks, improvements of this magnitude are typically regarded as substantial.

---

> > > ### Author Response · Authors · 2025-11-21
> > >
> > > **W7. Lack of GT in Real Images:**
> > >
> > > > *Fig. 7 shows real-image results but without ground truth, making it impossible to assess accuracy. The kernels estimated for real images (e.g., DSLR without stabilization) appear unstructured—are they physically plausible?*
> > >
> > > For truly real images, the main challenge is the absence of ground-truth HR images and downscaling kernels. Consequently, evaluation must rely on qualitative visual comparisons. Fig. 7 shows real images collected from diverse sources. We observe that our method recovers finer structures and readable text that competing blind SR methods fail to reconstruct.  We further encourage the reviewer to consult the appendix, where we provide additional real image examples.
> > >
> > > Regarding the shape of the estimated kernels, real camera PSFs are not perfectly structured Gaussians. They result from a combination of hand tremor, small scene motions, and in-camera processing, and therefore often appear irregular and non-Gaussian. The kernels we estimate exhibit exactly this behavior and, when used within our reconstruction pipeline, lead to sharper edges and more legible details than competing methods. This indicates that the estimated kernels are physically plausible and useful, even if they do not resemble idealized Gaussian models.

---

> > > > ### Comment · Reviewer_cUZR · 2025-11-27
> > > > **rebuttal disscussion**
> > > >
> > > > Thanks for the clarification from the authors. I've reviewed the response, but I still have significant doubts about the "Synthetic Stress-Test Kernels". Relying on kernels that don't mirror any real-case scenario to claim superiority raises questions about the practical relevance of the method. Additionally, this performance comes at a steep price in terms of computational efficiency. As my main concerns haven't been addressed, I'll be keeping my original score.

---

> > > > > ### Author Response · Authors · 2025-11-27
> > > > >
> > > > > We thank the reviewer for the follow-up. We would like to clarify the following:
> > > > >
> > > > > - “Synthetic Stress-Test Kernel”:
> > > > > \
> > > > > Please note that the evaluation of our method does not solely rely on stress test kernels, but **also consists of real motion kernels and real-world images**. We show that other algorithms consistently fail in such cases, while KernelFusion succeeds in performing superior SR and retrieving the corresponding kernel. Specifically, the results shown on real images point out the practical relevance of our method.
> > > > >
> > > > > - Computational Efficiency:
> > > > > \
> > > > > The goal of this paper is to **establish feasibility**, and not to establish a commercial SR system in terms of inference time. In addition, we provided a suggestion and the experimental data on how to improve the training time.
> > > > >
> > > > > We believe we addressed all the reviewer’s questions and remarks in full detail, as requested by the reviewer. We will be happy if the reviewer could relate to all our answers above, and point out what is still missing.

---

> > > > > > ### Comment · Reviewer_cUZR · 2025-11-28
> > > > > > **discussion**
> > > > > >
> > > > > > Thank you for your follow up comments. I feel that the authors may not have fully understood my main concerns. My critique does not focus on whether real world scenarios were validated, but rather on the paper’s central contribution—the design underlying the synthetic stress test kernel estimation. This approach results in a model that is highly time-consuming and computationally complex. In my opinion, the logical foundation for addressing the blind super-resolution problem in this way is fundamentally problematic.
> > > > > >
> > > > > > A superior method should indeed advance beyond existing approaches, which is a worthwhile goal. However, this requires two key elements: first, demonstrating state-of-the-art performance on standard datasets and scenarios, which I am willing to accept based on the current results; and second, proving the method’s effectiveness in a specifically challenging case where existing methods fail. This paper, however, concentrates heavily on synthetic stress-test kernel estimation, which introduces unacceptable computational cost.
> > > > > >
> > > > > > The authors present real-world deblurring and super-resolution simulations almost incidentally, while describing the paper as a validation of feasibility that should not be judged on efficiency. This appears to be a substitution of concepts. The feasibility being established is that of synthetic kernel estimation, not real-world deblurring performance, and it is this very focus that leads to the extreme computational burden.
> > > > > >
> > > > > > I cannot raise my score until this core issue is addressed. I believe the paper requires substantial revision to resolve this fundamental technical weakness. Please understand that my detailed comments are offered in a constructive spirit. My goal is not to prevent acceptance, but to encourage reflection on how to design and present a well motivated and logically organized paper.
> > > > > >
> > > > > > To summarize, the paper’s emphasis on estimating synthetic and extreme kernels leads to prohibitive computational complexity. Although tests on real images show some effect, the main contribution lacks a clear logical or technical connection to real-world performance. It is as though immense effort has been invested in solving a problem that does not practically exist, at the cost of practical applicability. I hope the authors can reconsider the core design and narrative structure, and I look forward to a future version that better aligns motivation, technical contribution, and application.

---

> > > > > > > ### Author Response · Authors · 2025-12-02
> > > > > > >
> > > > > > > Thank you for your follow-up and for clearly explaining your perspective.
> > > > > > >
> > > > > > > We respectfully point out that there seems to be a **factual misunderstanding** regarding the nature of the data used in our paper. While we understand the concern regarding "synthetic" tasks, we wish to clarify that our work focuses on **real-world degradations**:
> > > > > > >
> > > > > > > - **Real-World Empirical Kernels:** The primary non-Gaussian benchmark in our paper (DIV2KFK) does not use synthetic kernels. It uses the Levin et al. (2009) dataset, which consists of empirically measured kernels extracted from real cameras. These capture authentic hand tremor and optical aberrations found in the physical world.
> > > > > > > - **Real Images:** We further demonstrate results on real-world photographs (Fig. 7, 11-18) where no ground truth exists, recovering legible text where SOTA methods fail.
> > > > > > > - **Synthetic Stress-Tests:** The synthetic shapes (L-shape, square) are used only in the Blind144 dataset as boundary stress-tests to diagnose the method's flexibility. They are diagnostic tools, not the target application .
> > > > > > >
> > > > > > > We hope this clarifies that we are not substituting concepts, but rather solving the problem of blind SR on **real-world, non-Gaussian camera kernels** (Levin et al.). The computational cost is the necessary trade-off for removing the inductive biases that limit other methods on these complex real-world kernels.

---

### Official Review · Reviewer_TT8b · 2025-10-30

**Soundness:** 3
**Presentation:** 4
**Contribution:** 2
**Rating:** 4
**Confidence:** 4

**Summary:**

This work proposes KernelFusion, which primarily focuses on accurate estimation of non-restricted SR kernels to effectively tackle blind SR problems. In the first phase of KernelFusion, it learns local patch distribution of a single LR image with a zero-shot PatchDiffusion (PD) model, by optimizing PD at test time. After PD is sufficiently trained, it is frozen and acts as a strong prior for patch similarity across different scales. In the second phase, the SR result and according SR kernel is simultaneously estimated, with each an UNet and an INN. The method effectively captures extreme non-gaussian SR kernels, and outperforms baselines under various blind SR settings.

**Strengths:**

- The overall writing is very clear and easy to follow, with well supported citations.
- The core idea is both intuitive and sound.
- The performance gain is significant for unrestricted complex kernels (e.g., Blind144, DIV2KFK).
- The method does not require pretraining, and thus, can adapt to arbitrary kernels.

**Weaknesses:**

The reviewer sincerely appreciates the authors’ efforts in this work. In its current state, I lean toward a borderline reject due to the weaknesses below. However, **I am willing to revise my evaluation to accept if these concerns are adequately addressed**. Please, counter-argue about my concerns and provide according experimental results.

---

**Weakness 1: Limited performance on conventional kernels**

In Table 2, the performance gain of **KernelFusion** is significant for *Blind144* and *DIV2K-FancyKernels (DIV2KFK)*. Since these test sets include extreme kernels (e.g., K0, K1, K2 in Fig. 5), these results align well with the aim of the paper: accurately estimating extreme **unrestricted** kernels.

However, **KernelFusion** shows limited performance on *DIV2K-RK*, which uses random Gaussian kernels. Considering that such kernels are more common (and thus more practical), I have concerns about the performance of **KernelFusion** in real-world use cases. Because **blind SR** primarily focuses on practical scenarios (as bicubic SR tasks), this limitation is critical.

---

**Weakness 2: Missing analysis for the choice of INR**

> *“We suspect that this limitation stems from the implicit bias of the CNN and MLP architectures (Line 105).”*

One of the key contributions of this work lies in the use of **INR architectures** to estimate complex non-Gaussian kernels. However, an analysis justifying this architectural choice is missing. It is necessary to include both (1) the final SR performance and (2) the kernel estimation results under alternative architectural designs for kernel estimation.

The reviewer suggests reporting SR performance and kernel estimation results

* for the extreme kernels K0, K1, and K2 (shown in Fig. 5), and
* under various architectural configurations, including (1) direct kernel optimization and (2) DIP-style architectures.

---

**Weakness 3: Lack of comparison with diffusion-based kernel estimation methods**

Accurately estimating complex kernels with diffusion models **without any prior (i.e., pretraining)** is an interesting aspect of this work. However, kernel estimation using diffusion models is **not entirely new**. Prior diffusion-based works are not sufficiently discussed in this paper.

For instance, **BlindDPS [1]** employs diffusion models to estimate non-linear kernels for solving arbitrary blind inverse problems. The reviewer highlights the need to compare and discuss this work in two respects:
* Although blind **SR** is not directly addressed in *BlindDPS*, extending it to SR is straightforward, as it only requires an additional subsampling step.
* The central objective of both works is accurate kernel estimation using diffusion models.

The reviewer recommends comparing **KernelFusion** with *BlindDPS* (and possibly other related works), both theoretically and experimentally.

[1] Chung, Hyungjin, et al. “Parallel diffusion models of operator and image for blind inverse problems.” *Proceedings of the IEEE/CVF Conference on Computer Vision and Pattern Recognition*, 2023.

**Questions:**

**Question 1: Confusing experimental setting**

The experimental configuration of **Ablation (iii, PD+UNet)** is confusing.
Does *“acting as a denoiser”* indicate that the UNet is used only once (before the reverse diffusion step by PD)?
In other words, is this configuration equivalent to using **line 15 of Algorithm 1** only, while skipping **line 20 of Algorithm 1**?

If this interpretation is correct, the reviewer suggests adding another ablation experiment: skipping **line 15** while *not* skipping **line 20**. This experiment would help validate the necessity of applying the UNet twice, while also addressing the concern mentioned below:

> “Due to the small receptive field, the network does not know global structure, effectively destroying it …” (Line 430)

---

**Question 2: Necessity of using a shared UNet**

KernelFusion employs the same UNet twice (once before and once after PD).
Please discuss (and experimentally compare if possible) a configuration that uses independent network weights for each UNet step.

---

**Question 3: KernelFusion is slow**

> *"Our aim is not to deliver a production-ready Blind-SR system, but to establish feasibility of unrestricted kernel estimation ... "* (Line 123)

KernelFusion currently requires approximately **20 minutes** to perform SR for a single image, which remains excessively slow in practice: a clear limitation. However, the reviewer agrees with the academic necessity of analyzing unrestricted kernel estimation (despite being slow) and also considers the statement above fair.

However, please discuss (and experimentally compare if possible) configurations that reduce the number of diffusion steps (or explore techniques designed to accelerate diffusion processes).

**Details Of Ethics Concerns:**

No ethical concerns.

---

> ### Author Response · Authors · 2025-11-21
>
> We thank the reviewer for the detailed and constructive assessment. We addressed the raised weaknesses (W) and questions (Q), supported by additional experimental data.
>
> ***
> **W1. Limited performance on conventional kernels:**
>
> As correctly pointed out by the reviewer and shown in Table 2 in the paper, while KernelFusion leads on DIV2KFK and Blind144 datasets, it is not the leader on DIV2KRK. However, we would like to emphasize two key points:
>
> (i) As mentioned in "Empirical Evaluation of SR” in Section 5, the leading methods in Table 2 evaluated on DIV2KRK have been tailored to the Gaussian setup. Our method is general and has no such assumptions.
> \
> (ii) Our method shows promising results on real-world examples (see Fig. 7 and examples in Appendix). Kernels in real-world examples can sometimes appear approximately Gaussian (see e.g. bottom example of Fig. 7), but they are often unstructured and motion like (first 3 examples of Fig. 7). In these cases, our method is the only one that recovers clearly legible text and fine structures, whereas competing Blind SR methods fail and introduce artifacts.
>
> ***
> **W2. Missing analysis for the choice of INR:**
> > *The reviewer suggests reporting SR performance and kernel estimation results*
> > - *for the extreme kernels K0, K1, and K2 (shown in Fig. 5), and*
> > - *under various architectural configurations, including (1) direct kernel optimization and (2) DIP-style architectures.*
>
> We agree that our choice of INR for kernel estimation may benefit from a more thorough justification. As suggested by the reviewer, we now added a kernel-network ablation experiment on the 3 extreme kernels K0, K1, K2 from Blind144. This new experiment is now included in Appendix. C.1 of the revised manuscript, and will be referred to from the paper. We ablate & compare 3 different kernel estimation approaches:
>
> - “Direct”: directly optimizing each kernel entry as a parameter (no network), within a predefined kernel-grid.
> - “CNN”: a shallow linear convolutional “kernel network” adapted from KernelGAN.
> - “INR”: our proposed implicit neural representation.
>
> We evaluate each of these architectures in two settings: (i) within the full KernelFusion framework, and (ii) in a DIP-style SR setup.
> The resulting SR PSNRs on K0-K2 are summarized in new Table 4 in the Appendix, and show superiority of our combined INR architecture with our original KernelFusion framework over the alternatives. Nevertheless, the direct optimization’s performance in the HR image PSNR is also quite good (although not as good as INR). Moreover, looking at the recovered kernels themselves (see new Fig. 19 and 20 in the Appendix), it is evident that the INR architecture recovers much more accurate kernels than the direct optimization.
>
> ***
> **W3. Lack of comparison with diffusion-based kernel estimation methods:**
>
> > *The reviewer recommends comparing KernelFusion with BlindDPS (and possibly other related works), both theoretically and experimentally.*
>
> Please note that we do compare against other diffusion based methods for Blind-SR (see Table 2 in the submitted paper). For example, DiffBIR [Lin et al. 2023], a blind restoration framework that leverages a diffusion prior. As to BlindDPS [Chung et al. 2023] referred to by the reviewer – this is indeed a very relevant work using diffusion priors for blind inverse problems (thanks for pointing out). They construct two diffusion models, one as a prior over images and one as a prior over the kernel, and run parallel reverse diffusion to jointly estimate both. For blind deblurring, they:
>
>
> - Train a score model over blur kernels on 60k synthetic kernels (motion + Gaussian) for millions of steps using a U-Net.
> - Train a separate image prior on large face/image datasets (FFHQ, AFHQ, ImageNet).
>
> While this approach is conceptually close to ours in spirit (diffusion prior for the operator), note that it is *NOT Zero-Shot, assumes a specific prior training-set, and operates in a very different regime*. In particular, BlindDPS explicitly trains its kernel diffusion model on motion and Gaussian kernels (e.g. 60k generated blur kernels of those types). As a result, its generalization is inherently tied to these kernel families. Extending BlindDPS to SR is theoretically possible by incorporating a subsampling operator into the forward model, but it would still require:
>
> 1. Pre-training a kernel diffusion model on a SR-specific kernel dataset.
> 2. Operating within the kernel distributions seen during training.
>
> We will cite this paper in the revised version, and explain those differences.

---

> > ### Author Response · Authors · 2025-11-21
> >
> > **Q1. Confusing experimental setting:**
> >
> > > *The experimental configuration of Ablation (iii, PD+UNet) is confusing. Does “acting as a denoiser” indicate that the UNet is used only once (before the reverse diffusion step by PD)? In other words, is this configuration equivalent to using line 15 of Algorithm 1 only, while skipping line 20 of Algorithm 1? … “Due to the small receptive field, the network does not know global structure, effectively destroying it …” (Line 430)*
> >
> > We would like to clarify Ablation (iii): The configuration “PD + UNET” refers to the naive implementation with a UNET. It corresponds to the setup skipping lines 15 to 17 in the algorithm: PD gets $x_t$ as input (which corresponds to a noisy input), predicts $v_\theta$, and then reconstructs $x_{0,t,\theta}$ (line 19). Afterwards the UNET is applied (line 20). This way, PD gets the input it has been trained on (“a noisy input”), while the UNET is applied on $x_{0,t,\theta}$ which corresponds to the image prediction at timestep t  (“the clean output”). We will clarify this in the revised paper.
> >
> > We would like to further clarify the “small receptive field”: Our patch diffusion (PD) model is trained with a small receptive field – i.e., the receptive field of each output pixel in PD is a 15x15 ‘theoretical neighborhood’ in the input image (which in practice means a much smaller neighborhood). Therefore, PD learns only **local** patch statistics, and has no explicit notion of global image layout. We now created a new Fig. 21 in the revised Appendix to illustrate this: when a small amount of noise is added (small t), the noisy image still preserves the global structure, and PD can recover a reasonable reconstruction. However, when strong noise (large t) is added, the global structure in the noisy image is completely destroyed. PD then reconstructs an image that matches the patch distribution of the LR image it was trained on, but with no coherent global arrangement, yielding a mosaic of local patches. This is why we complement PD with a UNet that acts as a global image prior. Further explanation can be found in the new Appendix F.
> >
> > ***
> > **Q2. Necessity of using a shared UNet**
> >
> > > *KernelFusion employs the same UNet twice (once before and once after PD). Please discuss (and experimentally compare if possible) a configuration that uses independent network weights for each UNet step.*
> >
> > KernelFusion uses the same UNet twice, once before and once after PD, both times at the respective prediction of the “clean target” $\hat{x}_{0}$. Alternatively, one could use 2 identical UNets that do not share weights and are optimized individually. We run an additional ablation using 2 separate UNets with identical configurations and find the following results:
> >
> > - KernelFusion (2 same UNet as paper): 27.191 dB PSNR
> > - KernelFusion (2 separate UNets): 26.498 dB PSNR
> >
> > Applying a single UNet twice is clearly of advantage compared to separate optimization. The intuition behind applying the same UNet twice can be described as follows: Once the target prediction is sufficiently good at small t’s, the UNet output difference between $x_{0}$, at $t+1$ (before PD) and $t$ (after PD) should be marginal as the same images will have the same patch distribution. On the other side, we initialize Phase 2 with a bicubic guess of for $x_{0}$. Due to the high noise level, PD destroys global structure while adjusting the patch distribution and the second application of the UNet receives a heavily corrupted input. During this early phase, the first application of the UNet effectively acts as a regularizer that leverages the bicubic guess to provide the global structure information. We have added the new results in Table 5 in the Appendix of the revised script.

---

> > > ### Author Response · Authors · 2025-11-21
> > >
> > > **Q3. KernelFusion is slow**
> > > > *… the reviewer agrees with the academic necessity of analyzing unrestricted kernel estimation (despite being slow) and also considers the statement above fair. However, please discuss (and experimentally compare if possible) configurations that reduce the number of diffusion steps (or explore techniques designed to accelerate diffusion processes).*
> > >
> > > Optimization is the logical next step. Inspired by your feedback, we found that reducing Patch-Diffusion training to just 100k steps captures the necessary statistics. This yields a $4\times$ speedup (~5 mins instead of 20 mins) with negligible performance loss ($\sim0.07$ dB). We now added these results in Sec. E / Table 6 of the revised manuscript, to explicitly illustrate this cost-performance trade off and to show that the current 20 minute training per image setting is not strictly required for strong performance.
> > > Having said that, we want to stress two points:
> > >
> > > **(i) Breaking Assumptions:** As shown in Fig. 1, SOTA methods rely on fixed priors (Gaussian/Motion) and collapse when these assumptions break. Our paper points out this general limitation and brings it to the attention of the community. We further show that KernelFusion succeeds in this "impossible" regime by extracting the prior directly from the LR test image. We note that other single-image / internal-learning methods  (e.g., KernelGAN, MLMC, DKP) also require per-image optimization and typically take several minutes per image to produce their SR output. Yet, none of these can handle arbitrary downscaling kernels (as shown in Fig. 4 and 8).
> > > \
> > > **(ii) Feasibility:** We see a parallel to early foundational works like NeRF or Deep Image Prior, which required significant computation to establish feasibility in new domains. By proving that unrestricted kernel recovery is solvable, KernelFusion opens a new, assumption-free paradigm. We hope that the community will build on these foundations to bridge the gap to real-time efficiency.

---

> ### Author Response · Authors · 2025-11-27
>
> Dear Reviewer TT8b,
>
> As the discussion period advances, we wanted to gently ensure you had a chance to see our updates, which directly address the specific requests in your review:
>
> - we added the missing analysis supported by new experiments regarding the INR choice
> - we included new ablations and an accompanying discussion on the necessity of using a shared UNet
> - we responded in detail to all of your questions
>
> We remain available for any further discussion and would be happy to provide additional clarifications or improvements if needed. Your comments were valuable and we are open to any further suggestions.

---

### Official Review · Reviewer_xCF5 · 2025-10-31

**Soundness:** 3
**Presentation:** 2
**Contribution:** 3
**Rating:** 6
**Confidence:** 4

**Summary:**

This paper proposes KernelFusion, a zero-shot blind super-resolution method that simultaneously estimates an image-specific degradation kernel and the high-resolution image using a patch-based diffusion model. It addresses the limitation of existing methods that fail on complex, non-Gaussian kernels by leveraging internal learning, avoiding out-of-distribution problems.

**Strengths:**

- The combination of internal patch learning via diffusion models with implicit neural representations for kernel estimation is novel and creative.

- Successfully handles complex, non-Gaussian kernels where state-of-the-art methods fail, pushing blind SR into a more assumption-free paradigm.

- Strong experimental results on both synthetic kernels and real-world images convincingly demonstrate the advantage.

**Weaknesses:**

1. **Computational Cost and Practicality:** The most significant weakness is the computational cost. A 20-minute training time per image on a high-end GPU (L40S) makes it impractical for any real-time or even interactive application. While the authors correctly state their goal is to establish feasibility, this limitation is severe and should be a central point of discussion. Comparisons to other methods should ideally mention their relative runtimes to provide context.

2. **Limitation to Global Kernels:** The method is built on the assumption of a single, globally uniform kernel. As noted in the limitations, this prevents it from handling spatially varying blur, which is a common and important problem in real-world scenarios (e.g., motion blur in parts of an image, lens aberrations). This restricts the scope of its applicability.

3. **Ablation on Kernel Estimation Component:** While the ablation study in Table 3 effectively dissects the contributions to the final HR reconstruction quality (PSNR), it does not explicitly ablate the kernel estimation component. How crucial is the INR architecture compared to a simpler parameterization? A quantitative analysis (e.g., using kernel PSNR or MSE) of the estimated kernels in the different ablated settings would strengthen the claim that the INR is key to recovering complex kernels.

4. **Comparison to Other Zero-Shot Methods:** The comparison to the two-step process of KernelGAN+ZSSR is excellent and shows a clear advantage. However, it would be even more compelling to see a comparison against other recent internal-learning or zero-shot SR methods that are not based on diffusion models, to better isolate the benefit of the proposed diffusion-based approach.

**Questions:**

see weaknesses

---

> ### Author Response · Authors · 2025-11-21
>
> We thank the reviewer for the thoughtful evaluation. Below we address the reviewers concerns point-by-point:
>
> ***
> **W1: Computational Cost and Practicality:**
>
> This is a valid concern. KernelFusion is not yet fast enough for commercial deployment, but it serves a different purpose: establishing the scientific feasibility of a task previously considered impossible; Blind-SR with unrestricted kernels.
>
> That said, optimization is the logical next step. Inspired by your feedback, we found that reducing Patch-Diffusion training to just 100k steps captures the necessary statistics. This yields a $4\times$ speedup (~5 minutes) with negligible performance loss ($\sim0.07$ dB). We now added these results in Sec. E / Table 6 of the revised manuscript, to explicitly illustrate this cost-performance trade off and to show that the current 20 minute training per image setting is not strictly required for strong performance.
> Having said that, we want to stress two points:
>
> **(i) Breaking Assumptions:** As shown in Fig. 1, SOTA methods rely on fixed priors (Gaussian/Motion) and collapse when these assumptions break. Our paper points out this general limitation and brings it to the attention of the community. We further show that KernelFusion succeeds in this "impossible" regime by extracting the prior directly from the LR test image. We note that other single-image / internal-learning methods  (e.g. KernelGAN, MLMC, DKP) also require per-image optimization and typically take several minutes per image to produce their SR output. Yet, none of these can handle arbitrary downscaling kernels (as shown in Fig. 4 and 8).
> \
> **(ii) Feasibility:** We see a parallel to early foundational works like NeRF or Deep Image Prior, which required significant computation to establish feasibility in new domains. By proving that unrestricted kernel recovery is solvable, KernelFusion opens a new, assumption-free paradigm. We hope that the community will build on these foundations to bridge the gap to real-time efficiency.
>
> ***
> **W2.  Limitation to Global Kernels:**
>
> We agree that handling spatially varying blur is an important direction, and is a current limitation of our paper (as explicitly mentioned in the limitations, Section 5). In this work, however, we follow the standard blind SR setting, where a single global kernel is assumed. This assumption is used by the vast majority of blind SR methods and further is often represented in the training data of supervised baselines (e.g., SwinIR), making it the natural choice for a fair comparison. Our focus in this paper is to address the fundamental question of dealing with arbitrary unrestricted global kernels, an ability that existing methods lack even under the global kernel assumption. Extending KernelFusion to spatially varying blur is an interesting direction for future work, but is beyond the scope of this paper.
>
> ***
> **W3. Ablation on Kernel Estimation Component:**
>
> We agree that our choice of INR for kernel estimation may benefit from a more thorough ablation. This point was also raised by Reviewer TT8b. To address this concern, we followed the experiment suggested by Reviewer TT8b. As requested, we now added a kernel-network ablation experiment on the 3 extreme kernels K0, K1, K2 from Blind144. This new experiment is now included in Appendix. C.1 of the revised manuscript, and will be referred to from the paper. We ablate & compare 3 different kernel estimation approaches:
>
> - “Direct”: directly optimizing each kernel entry as a parameter (no network), within a predefined kernel-grid.
> - “CNN”: a shallow linear convolutional “kernel network” adapted from KernelGAN.
> - “INR”: our proposed implicit neural representation.
>
> We evaluate each of these architectures in two settings: (i) within the full KernelFusion framework, and (ii) in a DIP-style SR setup.
> The resulting SR PSNRs on K0-K2 are summarized in new Table 4 in the Appendix, and show superiority of our combined INR architecture with our original KernelFusion framework over the alternatives. Nevertheless, the direct optimization’s performance in the HR image PSNR is also quite good (although not as good as INR). Moreover, looking at the recovered kernels themselves (see new Fig. 19 and 20 in the Appendix), it is evident that the INR architecture recovers much more accurate kernels than the direct optimization.
>
> ***
> **W4. Comparison to Other Zero-Shot Methods:**
>
> We would like to draw the reviewers attention to Table 2 that includes several internal learning / zero-shot baselines that are not diffusion based: KernelGAN + ZSSR, MLMC and DKP. All these methods are single-image and internal-learning approaches, that rely only on the input LR image (no external paired training data) and therefore are exactly in the regime the reviewer refers to. KernelFusion consistently outperforms all of them on all three datasets (Table 2), both in terms of SR quality and the estimated kernel accuracy. See Fig. 4 and Fig. 6 for qualitative comparison.

---

> > ### Comment · Reviewer_xCF5 · 2025-11-27
> >
> > I have read the reviewer's rebuttal, which addressed my concern a lot. I raised my score to 8.

---

### Author Response · Authors · 2025-11-21

We thank the reviewers for the thoughtful and constructive feedback. We address the weaknesses (W) and questions (Q) raised by the reviewers point-by-point in the provided rebuttal boxes for each review individually (supported by additional experimental data). We uploaded a revised script with the changes and additions marked in red. A brief summary follows:

- **Improving the runtime of PD** by training PD for 100K steps delivers comparable results while improving training time **by a factor of 4**.
- The added **Kernel Network ablation** shows the superior performance of the INR network architecture compared to other architectural choices
- We enriched our ablation study to cover additional aspects such as why **applying the same UNet twice** is better than training two individual UNets with non-shared weights
- Intuition and visual examples are provided on limited **receptive field** of patch diffusion
- Added the comparison to an additional competitor in Table 2: Real-ESRGAN
- Fixed minor issues pointed by the reviewers

We appreciate the reviewer's time and hope the revised manuscript addresses all concerns.

---

### Meta-Review · Area_Chair_nrgm · 2025-12-24

**Summary:**

This paper proposes to solve blind super-resolution with complex downscaling degradations. To achieve this, it proposes to use patch recurrence across scales to model the patch distribution from LR image based on diffusion. With this distribution, they can recover HR image with the blur kernel simultaneously. Even though only one reviewer accepts this paper, I still believe the idea is meaningful and can motivate the SR community. The authors should carefully revise the paper according to the content of the rebuttal in the final version.

**Reviewer Concerns:**

1. For the computation cost (xCF5 TT8b t2tb). Most of the reviewers point it out. However, I think the proposed idea is interesting, useful and well-motivated. And accurately estimating kernels is indeed important for blind super-resolution. From this perspective, this paper has the potential to motivate the community.
2. For the ablation of kernel estimation (xCF5 TT8b). The authors have added some experiments and xCF5 is satisfied with it.
3. For the unclear of paper writing (TT8b t2tb). The authors have clarified it in the rebuttal. And it should be added to the final version.
4. For the limitation to global kernels (xCF5). The authors point out that they focus on arbitrary unrestricted global kernels other than non

**Reviewer Scores:**

From my perspective, TT8b might change his mind and accept this paper. As to t2tb, this remaining major concern is the computation costs and clarity of the method. I agree that the proposed method is not efficient. However, this paper proposes an interesting method to more accurately estimate the complex blur kernel and then restore the HR image, which can motivate the community. For the clarity of the method, the authors can clarify them in the revised version.

---

### Decision · Program_Chairs · 2026-01-26

Accept (Poster)